# 1S-DAug: One-Shot Data Augmentation for Robust Few-Shot Generalization

## Abstract

Few-shot learning (FSL) demands generalization to novel classes based on just a few shots of labeled examples, a setting where traditional test-time augmentations fail to be effective. We introduce 1S-DAug, a one-shot generative augmentation operator that synthesizes diverse yet faithful variants from just one example image at test time. 1S-DAug couples traditional geometric perturbations with controlled noise injection and a denoising diffusion process conditioned on the original image. The generated images are then encoded and aggregated, alongside the original image, into a combined representation for more robust FSL predictions. Integrated as a training-free model-agnostic plugin, 1S-DAug consistently improves FSL across standard benchmarks of 4 different datasets without any model parameter update, including achieving 10% proportional accuracy improvement on the miniImagenet 5-way-1-shot benchmark. Codes will be released.

## 1 Introduction

Few-shot learning (FSL) is important for recognition systems deployed in the wild. While deep neural networks attain strong performance given abundant supervision, their accuracy degrades in rare-case generalization (Wang et al., 2020). Real-world data are long-tailed; rare categories with limited labels persist and cap overall system performance even as head classes continue to grow (Kang et al., 2020). Scarcity in the target domain induces a train-test gap that manifests as high generalization error on novel classes at test time (Wang et al., 2020).

FSL is a concrete instance of this long-tail problem. The FSL model must assign labels to previously unseen classes using only a handful of labeled examples per class (Vinyals et al., 2016; Snell et al., 2017). This low-data regime appears in practical settings such as medical imaging for rare diseases and autonomous driving with open-world, unpredictable events (Liu & Feng, 2024). The central challenge is to achieve robust generalization under strict label scarcity and distribution shift.

Data augmentation offers a natural handle on this challenge (Dvornik et al., 2019; Zhou, 2012). From an ensemble perspective, test-time augmentation aggregates predictions across multiple views of the same input and averages out error; in an idealized independence thought experiment, a single-view error rate $\varepsilon$ would combine as $\varepsilon^m$ for $m$ views, while in practice correlation attenuates but does not eliminate the benefit (Zhou, 2012). From a margin perspective, classical generalization bounds can relate test error to data radius; augmentations that contract effective data radii can lower the Rademacher complexity and tighten such bounds (Bartlett & Mendelson, 2002; Bai et al., 2025). On the training side, augmentation increases observed data amount and can tighten the generalization bound further (Hariharan & Girshick, 2017b; Schwartz et al., 2018).

However, the augmentation is effective only with both diversity and accuracy (Zhou, 2012). Standard geometric or photometric transforms like cropping, resizing and scaling often add limited new information and may degrade image quality (Bai et al., 2025). In FSL, where each example carries high influence, degradations are especially harmful at test time, and the model must rely on precise visual cues. Achieving high diversity while preserving class-defining content is therefore central.

Generative data augmentation has potential, but deploying it under FSL constraints is nontrivial. Image-to-image translation with adversarial training (e.g., FUNIT, CycleGAN) can be effective in specific domains, yet it is prone to training instabilities and to inconsistent quality across dissimilar object categories and poses (Liu et al., 2019a; Zhu et al., 2017). Prior attempts to employ GANs

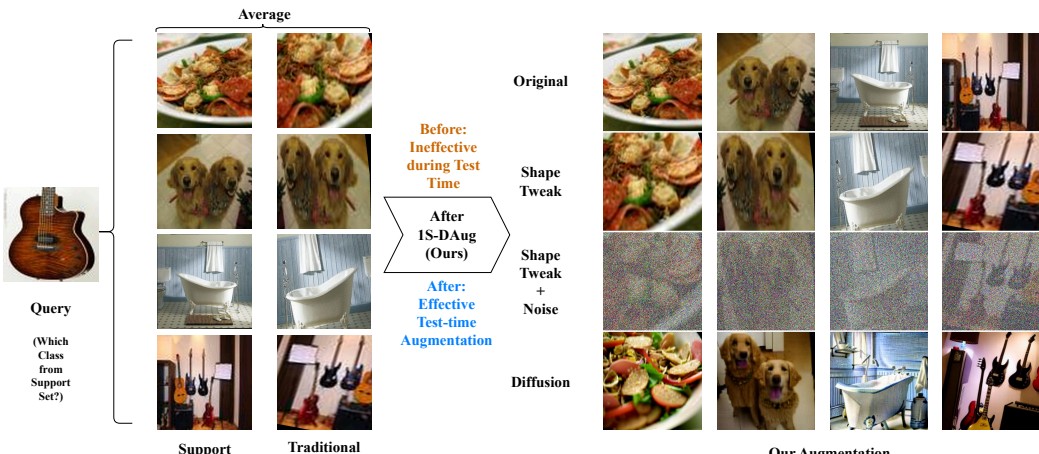

Figure 1: Pipeline of our one-shot, test-time augmentation. Given a query and supports, we apply a shape tweak and controlled noise, then perform attention-conditioned diffusion to synthesize class-faithful variants. Features from the original and the generated views are averaged before the few-shot head. For contrast, traditional test-time geometric transformations provide limited diversity.

for FSL augmentation typically restrict themselves to training-time augmentation (Hariharan & Girshick, 2017b; Schwartz et al., 2018; Hariharan & Girshick, 2017a). While generating data for seen classes is common, quality synthesis for unseen classes is challenging. To our best knowledge, Bai et al. (2025) is the only prior work explicitly using test-time generative augmentation for FSL (Liang et al., 2024), which relies on gan-based image-to-image translation comprising two samples and focuses on animal faces instead of more-diverse objects, limited in the broad dataset applicability (Liu et al., 2019a). A few recent work leverages diffusion-generated images for training data augmentation. However, the methods rely on fine-tuning using text prompts or additional target-class samples (Trabucco et al., 2024; He et al., 2023), thus not suitable for the challenging FSL test-time set-up.

We propose *1S-DAug* (One-Shot Data Augmentation), a one-shot generative operator that, given a single input image at test time, synthesizes a set of diverse yet faithful variants. The operator combines three ingredients: (i) class-preserving shape perturbations to broaden pose and layout coverage; (ii) controlled noise injection to hide non-class-defining and distorted details; and (iii) denoising guided by attention-based conditioning on the original image so that content-defining cues are preserved while appearance and pose vary in a controlled manner. During few-shot classification, the generated variants are processed by a fixed encoder and aggregated with the original into a single representation. The method is model-agnostic and requires no retraining. As large pretrained models proliferate and model sizes continue to explode, fine-tuning or model-side ensemble can be burdensome or infeasible due to compute or **restricted parameter access** (Zheng et al., 2023). A test-time, data-side plugin that treats the predictor in model-agnostic ways is thus attractive. In experiments on four FSL benchmarks (e.g., miniImagenet 5-way-1-shot), our method yields consistent gains without modifying any model parameters. Our contributions are as follows:

- We introduce *1S-DAug*, a one-shot data augmentation method. Given a single image, it leverages attention-guided diffusion with controlled noise to generate faithful yet diverse variants suitable for inference on unseen classes. Though synthetic data augmentation with target-class supervision is not new, to our best knowledge, we are the **first** to abide by the strict one-shot set-up for synthetic data augmentation, not relying on any label.
- Since our augmentation is strictly one-shot during inference, it can fulfill the challenging FSL test-time augmentation requirement. As FSL test-time augmentation is practically meaningful but **underexplored**, we analyse 1S-DAug's integration with FSL, where diversity and accuracy should be both achieved for performance optimality.
- We implement 1S-DAug as a test-time plugin for trained FSL models, and we evaluate on benchmarks of four datasets, observing **consistent gains** (e.g., up to 20% relative improvement on 5-way-1-shot benchmarks) without any underlying parameter update to the off-the-shelf models.

## 2 RELATED WORK

For **few-shot learning**, augmentation-based approaches expand training diversity via feature hallucination (Hariharan & Girshick, 2017a) or GAN-driven synthesis (Wang et al., 2018), but they are largely *training-time* and rely on supervision from base classes. In contrast, *test-time* augmentation for FSL is harder; it must generate high-quality, class-faithful variants for unseen classes without any retraining or labels. The only explicit work in this direction is FSL-Rectifier (Bai et al., 2025), which uses FUNIT (Liu et al., 2019a) to combine the shape of one image with the class-defining style of another. This serves as a proof of concept, mainly targeting animal-face datasets with limited broad applicability. For **diffusion models**, attention-based conditioning adapters now inject external signals (e.g., an input image) into cross-attention, enabling faithful, controllable generation (Mou et al., 2023). SDEdit (Meng et al., 2022) adds controlled noise and then denoises with conditioning. However, lower noise level yields minimal change, and the higher sacrifices faithfulness (e.g., change object type), thus not suitable as data augmentation that requires both diversity and faithfulness. There are a few recent works using the diffusion-based synthetic images for training augmentation outside of few-shot learning, but they rely on fine-tuning based on text prompts or additional test-class samples, and are thus limited for test-time augmentation (He et al., 2023; Benigmim et al., 2023; Trabucco et al., 2024). More related works are available in Appendix Section §D.

## 3 DIFFUSION PRELIMINARIES

Let $x_0 \in \mathcal{X}$ be an image and $x_t$ its noised version at step $t \in \{1, \ldots, T\}$. A variance schedule $(\beta_t)_{t=1}^T$ defines $\alpha_t = 1 - \beta_t$ and $\bar{\alpha}_t = \prod_{\tau=1}^t \alpha_\tau$. We use the variance-preserving (VP) forward process (Sohl-Dickstein et al., 2015; Ho et al., 2020; Nichol & Dhariwal, 2021; Meng et al., 2022)

$$q(x_t \mid x_0) = \mathcal{N}\big(\sqrt{\bar{\alpha}_t}\, x_0,\ (1 - \bar{\alpha}_t)\mathbf{I}\big), \tag{1}$$

so that sampling $x_t$ can be written as $x_t = \sqrt{\bar{\alpha}_t}\, x_0 + \sqrt{1 - \bar{\alpha}_t}\, \epsilon$ with $\epsilon \sim \mathcal{N}(0, \mathbf{I})$.

A user noise level $\eta \in [0, 1]$ is mapped to a discrete start time $t_0 \in \{1, \ldots, T\}$ by matching cumulative noise:

$$t_0 = \arg\min_{t \in \{1, \ldots, T\}} \big|(1 - \bar{\alpha}_t) - \eta^2\big|. \tag{2}$$

Let $\hat{\epsilon}_\varphi$ be a learned noise predictor with parameters $\varphi$, and let $c_t$ be the conditioning signal at time $t$. The VP reverse update is

$$x_{t-1} = \mu_\varphi(x_t, t, c_t) + \sigma_t \epsilon, \tag{3}$$

with $\epsilon \sim \mathcal{N}(0, \mathbf{I})$ and sampler noise level $\sigma_t$ (we use $\sigma_t=0$ in our experiments), where $\mu_\varphi(x_t, t, c_t) = \alpha_t^{-1/2}\big(x_t - \frac{1-\alpha_t}{\sqrt{1-\bar{\alpha}_t}}\, \hat{\epsilon}_\varphi(x_t, t, c_t)\big)$. The same formulas apply in a latent space $z_t$ via an encoder–decoder $(\mathrm{Enc}, \mathrm{Dec})$, with denoising output $\tilde{x} = \mathrm{Dec}(z_0)$ (Rombach et al., 2022b).

## 4 METHOD

**FSL Problem Setup.**   We operate in standard inductive FSL classification, where a trained encoder $\Phi_\theta : \mathcal{X} \to \mathbb{R}^d$ and a non-parametric classifier are evaluated on novel classes. Given a support set of labelled examples and an unlabelled query, the model must decide which support class the query belongs to. Typical FSL methods compute similarities between the query representation and the representations of support examples under a distance or similarity metric, and assign the query to the closest class prototype Wang et al. (2020); Sung et al. (2018); Ye et al. (2020); Snell et al. (2017). In what follows we denote the encoder by $\Phi_\theta$ but, when $\theta$ is fixed, we often simply write $\Phi$ for brevity. Our goal is to wrap any such trained FSL model with a training-free test-time augmentation operator that, given a single image, synthesises faithful yet diverse variants and aggregates their representations for prediction.

**1S-DAug.**   We produce variants of a single image by (1) applying traditional geometric changes, (2) injecting a controlled amount of noise to determine edit magnitude, and (3) denoising via a

diffusion process conditioned on the source image so that content-defining attributes are preserved while details and pose can vary. We write the resulting single-image augmentation operator as

$$\mathcal{A}(x; \upsilon) = \text{Den}_\varphi\Big(\text{Noi}_\eta\big(T_\psi(x)\big), \lambda_{\text{img}}\Big) \in \mathcal{X}, \tag{4}$$

where $\upsilon = (\psi, \eta, \lambda_{\text{img}})$ collects the geometric, noising, and conditioning hyperparameters; $T_\psi$ is a sampled geometric transform; $\text{Noi}_\eta$ (add noise) applies the forward diffusion process up to a start time $t_0$ determined by the noise level $\eta$ (e.g. via equation 2); and $\text{Den}_\varphi$ (denoise), parametrized by fixed diffusion parameters $\varphi$, runs the reverse process from $t_0$ to 0 with image-conditioned attention. We next detail each stage of this operator.

**Stage 1: Shape Tweak.** Changes in pose/layout increase coverage of plausible views without altering class identity. Let $\psi$ be shape-tweak parameters and

$$x_{\text{geom}} = T_\psi(x),$$

where $T_\psi$ is drawn from a family of traditional image transformations, composed as rotations, anisotropic stretches, translations, perspective jitters, and horizontal flips.

**Stage 2: Controlled Noising.** The noise level sets the change magnitude during the diffusion denoising pipeline. Lower noise emphasises faithfulness; higher noise hides geometric distortion better and yields more diversity. We use the variance-preserving (VP) forward kernel (Sohl-Dickstein et al., 2015; Ho et al., 2020; Nichol & Dhariwal, 2021; Meng et al., 2022). Let $t \in \{1, \ldots, T\}$ index discrete diffusion steps, let $\beta_t \in (0, 1)$ be a variance schedule, and define $\alpha_t := 1 - \beta_t$ and $\bar{\alpha}_t := \prod_{\tau=1}^{t} \alpha_\tau$. Given a user noise level $\eta \in [0, 1]$, we choose a start time $t_0 = t(\eta)$ (e.g., by matching cumulative noise as in equation 2), and sample

$$x_{t_0} \sim q(x_{t_0} \mid x_{\text{geom}}) = \mathcal{N}\Big(\sqrt{\bar{\alpha}_{t_0}}\, x_{\text{geom}}, \, (1 - \bar{\alpha}_{t_0})\, \mathbf{I}\Big), \tag{5}$$

where $x_{\text{geom}}$ is the geometrically perturbed input (Stage 1), $\mathbf{I}$ is the identity covariance, and $\mathcal{N}$ denotes a Gaussian distribution.

**Stage 3: Image-Conditioned Diffusion Denoising.** Let $z_t$ be the latent at time $t$, and let $f_{\text{img}}(x) \in \mathbb{R}^{L \times d_k}$ and $f_{\text{txt}}(p) \in \mathbb{R}^{M \times d_k}$ be fixed-encoder outputs for the condition image $x$ and optional text prompt $p$. For a U-Net (Ronneberger et al., 2015) block at time $t$, with queries $Q_t = W_Q z_t \in \mathbb{R}^{N_q \times d_k}$ (here $W_Q$ denotes the query weights and biases and $N_q$ is the number of query tokens) and keys/values $K_t, V_t \in \mathbb{R}^{(M+L) \times d_k}$, the cross-attention (Vaswani et al., 2017) is

$$A_t(Q_t, K_t, V_t) = \text{softmax}\Big(\frac{Q_t K_t^\top}{\sqrt{d_k}}\Big) V_t, \tag{6}$$

and we concatenate text/image tokens with a scalar weight $\lambda_{\text{img}} \geq 0$:

$$K_t = \big[K_t^{\text{txt}}, \, \lambda_{\text{img}} K_t^{\text{img}}\big], \quad V_t = \big[V_t^{\text{txt}}, \, \lambda_{\text{img}} V_t^{\text{img}}\big], \tag{7}$$

with $K_t^{\text{txt}}, V_t^{\text{txt}} = W_{\text{txt},t} f_{\text{txt}}(p)$ and $K_t^{\text{img}}, V_t^{\text{img}} = W_{\text{img},t} f_{\text{img}}(x)$. We set the conditioning variable for the reverse update to $c_t := A_t(Q_t, K_t, V_t)$. The VP reverse step equation 3 then reads

$$z_{t-1} = \mu_\varphi(z_t, t, c_t) + \sigma_t \epsilon, \qquad \epsilon \sim \mathcal{N}(0, \mathbf{I}), \tag{8}$$

and rolling out $t_0 \to 0$ produces $\tilde{x} = \text{Dec}(z_0)$.

**FSL Feature Aggregation.** Averaging features over faithful but non-identical views draws representations toward class-typical regions and improves robustness for non-parametric few-shot classifiers (Snell et al., 2017; Chen et al., 2019; Ye et al., 2020; Bai et al., 2025). For each image $x$ we generate $K_a^{\text{sup}} + 1$ support views $\tilde{x}^{(k)} = \mathcal{A}(x; \upsilon^{(k)})$ with $\tilde{x}^{(0)} \equiv x$, where each $\upsilon^{(k)}$ denotes a fresh sample of augmentation hyperparameters, and form an aggregated support embedding

$$\bar{z}_{\text{sup}}(x) = \sum_{k=0}^{K_a^{\text{sup}}} \alpha_k^{\text{sup}}\, \Phi\big(\tilde{x}^{(k)}\big), \qquad \alpha_k^{\text{sup}} \geq 0, \, \sum_{k=0}^{K_a^{\text{sup}}} \alpha_k^{\text{sup}} = 1. \tag{9}$$

Given a support set $S = \{(s_i, y_i)\}_{i=1}^{NK}$ with $N$ classes and $K$ shots per class, class prototypes are $p_c = \frac{1}{K} \sum_{i:\, y_i = c} \bar{z}_{\text{sup}}(s_i)$. For each query $q$ we generate $K_a^{\text{qry}} + 1$ query views $\tilde{q}^{(k)} = \mathcal{A}(q; v^{(k)})$, encode them as $z^{(k)}(q) = \Phi(\tilde{q}^{(k)})$, and compute per-view logits $\ell_c^{(k)}(q) = \kappa\big(z^{(k)}(q), p_c\big)$ for a chosen similarity $\kappa$ (Euclidean or cosine). We then perform query-side logit averaging for FSL prediction:

$$\tilde{\ell}_c(q) := \sum_{k=0}^{K_a^{\text{qry}}} \alpha_k^{\text{qry}} \ell_c^{(k)}(q), \qquad \alpha_k^{\text{qry}} \geq 0, \ \sum_{k=0}^{K_a^{\text{qry}}} \alpha_k^{\text{qry}} = 1, \qquad \hat{y}(q) = \arg \max_{c \in \{1,\dots,N\}} \tilde{\ell}_c(q). \quad (10)$$

# 5 THEORETICAL INSIGHTS

We analyse 1S-DAug in the standard episodic few-shot setting with a single trainable encoder and a fixed Euclidean nearest-prototype classifier. We show that (i) a simple risk decomposition separates the ensemble benefit into accuracy and diversity, and (ii) a margin-based generalisation bound for the encoder becomes strictly tighter after augmentation, via both empirical margin and feature-radius reduction. Lastly, based on the generalisation bound, we compare training and test-time augmentation in Appendix Section §H.7, highlighting the latter's comparative advantage. All missing definitions and proofs are deferred to Appendix Sections §G, §H.2, §H.5, and §H.6.

## 5.1 EPISODIC EUCLIDEAN MODEL AND TEST-TIME AUGMENTATION

We consider the encoder-plus-Euclidean-prototype classifier of Section 4. In an $N$-way $K$-shot episode, class prototypes $p_c$ are formed by averaging support embeddings, and a query $q$ is assigned to the nearest prototype in squared Euclidean distance $\|\Phi_\theta(q) - p_c\|_2^2$. For the theory we reduce episodes to binary query–prototype pairs $x = (q, p)$ with label $y \in \{-1, 1\}$, write

$$\Omega_\theta(x) := \Phi_\theta(q) - p \quad \text{and} \quad g_\theta(x) := -\|\Omega_\theta(x)\|_2^2,$$

and assume a uniform radius bound $\|\Omega_\theta(x)\|_2 \leq r_0$. Under test-time augmentation, multiple query views are combined by logit averaging; for squared Euclidean scores this is equivalent to using an averaged query embedding and hence an aggregated difference feature $\bar{\Omega}_\theta(x)$. Full episodic details and the logit–feature equivalence are deferred to Appendix Section §H.1 and §H.3.

## 5.2 RISK DECOMPOSITION INTO ACCURACY AND DIVERSITY

We first quantify the ensemble effect at the pairwise level. For any real-valued predictor $g$ on pairs $x = (q, p)$ we use the scaled squared-loss risk

$$\mathcal{R}(g) := \frac{1}{4} \, \mathbb{E}_{(x,y) \sim D} \big[ (g(x) - y)^2 \big], \quad (11)$$

which coincides with 0–1 pairwise error when $g(x) \in \{-1, 1\}$. In particular, if $g_\theta$ is a sign-valued score, the pairwise misclassification risk $R_{\text{cls}}(\theta) := \mathbb{P}_{(x,y) \sim D}\big(y\, g_\theta(x) \leq 0\big)$ satisfies $R_{\text{cls}}(\theta) = \mathcal{R}(g_\theta)$. Let $f(x)$ and $f_A(x)$ be the sign predictors associated with the base and an augmented view, and define the (two-view) ensemble $\tilde{f}(x) := \frac{1}{2}\big(f(x) + f_A(x)\big) \in \{-1, 0, 1\}$. A direct calculation (Appendix Section §G) yields:

**Proposition 1** (Pairwise risk decomposition). *With $\mathcal{R}(\cdot)$ as in equation 11,*

$$\mathcal{R}(\tilde{f}) - \mathcal{R}(f) = \underbrace{\frac{1}{4}\big(\mathbb{E}[f(x)y] - \mathbb{E}[f_A(x)y]\big)}_{\text{accuracy gap}} + \underbrace{\frac{1}{8}\big(\mathbb{E}[f(x)f_A(x)] - 1\big)}_{\text{diversity term}}. \quad (12)$$

Thus improvements come from (i) maintaining or improving single-view accuracy, and (ii) making the augmented predictions sufficiently diverse on hard examples. This matches the empirical behaviour of 1S-DAug, which is designed to generate plausible but different-shape query views.

### 5.3 Encoder generalisation, radius reduction, and 1S-DAug

With the encoder as the only learnable component, our method tightens a margin-based generalisation bound. Let $\mathcal{G} := \{g_\theta : \theta \in \Theta\}$ be the score class induced by the encoder, and let $R_{\mathrm{cls}}(\theta) := \mathbb{P}_{(x,y) \sim D}\big(y\, g_\theta(x) \leq 0\big)$ denote the pairwise misclassification risk. For a sample $S = \{(x_i, y_i)\}_{i=1}^m$ and margin parameter $\rho > 0$, let $\widehat{R}_{S,\rho}(\theta)$ be the empirical $\rho$-margin loss for $g_\theta$, and let $\widehat{\mathfrak{R}}_S(\mathcal{G})$ denote the empirical Rademacher complexity of $\mathcal{G}$. Formal definitions are deferred to Appendix Section §H.2. A standard margin-based Rademacher argument gives the following.

**Theorem 1** (Encoder margin bound). *For any $\rho > 0$ and $\delta > 0$, with probability at least $1 - \delta$ over $S \sim D^m$, every encoder $\theta$ satisfies*

$$R_{\mathrm{cls}}(\theta) \;\leq\; \widehat{R}_{S,\rho}(\theta) + \frac{2}{\rho}\,\widehat{\mathfrak{R}}_S(\mathcal{G}) + \sqrt{\frac{\log(1/\delta)}{2m}}. \tag{13}$$

Suppose $\Phi_\theta$ is a feedforward network with 1-Lipschitz nonlinearities and layer spectral norms $\|W_\ell\|_2 \leq s_\ell$ such that $\prod_{\ell=1}^L s_\ell \leq L_{\mathrm{enc}}$. Then $g_\theta$ is Lipschitz in the difference feature with constant proportional to $L_{\mathrm{enc}} r_0$, where $r_0 := \sup_x \|\Omega_\theta(x)\|_2$ is the (pre-augmentation) feature radius from Section 5.1. Using standard spectral-norm bounds, Lemma 2 (Appendix Section §H.2) shows that $\widehat{\mathfrak{R}}_S(\mathcal{G}) \lesssim L_{\mathrm{enc}} \frac{r_0}{\sqrt{m}}$, so the complexity term in equation 13 scales linearly with the feature radius $r_0$. Let $\hat{r} := \sup_x \|\bar{\Omega}_\theta(x)\|_2$ be the radius after augmentation and $\widehat{R}_{\widehat{S},\rho}(\theta)$ the empirical margin loss computed with the aggregated sample $\widehat{S}$. By convexity, augmentation cannot increase the radius:

**Lemma 1** (Radius contraction under augmentation). *If $\|\Omega_\theta^{(k)}(x)\|_2 \leq r_0$ for all $x$ and views $k$, then $\hat{r} := \sup_x \|\bar{\Omega}_\theta(x)\|_2$ satisfies $\hat{r} \leq r_0$. Moreover, under a simple symmetric i.i.d. model for the views $\{\Omega_\theta^{(k)}(x)\}$, the maximal empirical radius strictly decreases with non-zero probability.*

Combining Theorem 1, Lemma 2, and Lemma 1, the encoder-only risk before test-time augmentation satisfies

$$R_{\mathrm{cls}}(\theta) \;\leq\; \widehat{R}_{S,\rho}(\theta) + \frac{2C_{\mathrm{enc}} L_{\mathrm{enc}}}{\rho}\,\frac{r_0}{\sqrt{m}} + \sqrt{\frac{\log(1/\delta)}{2m}}, \tag{14}$$

for some architecture-dependent constant $C_{\mathrm{enc}} > 0$. After test-time augmentation, the bound is the same except that $\widehat{R}_{S,\rho}(\theta)$ is replaced with $\widehat{R}_{\widehat{S},\rho}(\theta)$, and $r_0$ with $\hat{r}$. Since $\hat{r} = \alpha r_0$ for some $\alpha \in (0, 1]$, the complexity term contracts by factor $\alpha$, and Appendix Section §H.5 further shows that augmentation tends to move representations towards their class prototypes, increasing margins and thereby reducing the empirical term as well (i.e., $\widehat{R}_{\widehat{S},\rho}(\theta) < \widehat{R}_{S,\rho}(\theta)$). In summary, 1S-DAug tightens the encoder's generalisation bound through both empirical margin and feature-radius reduction.

## 6 Experiments

### 6.1 Set-up and Main Results

**Datasets and Pre-processing.** We follow the standard 5-way-1/5-shot episodic evaluation on miniImagenet and tieredImagenet (Russakovsky et al., 2015); additional experiments are conducted on CUB (fine-grained birds) (Wah et al., 2011) and an animal-face dataset used previously for test-time augmentation studies (Animals) (Liu et al., 2019a; Bai et al., 2025). We adhere to the conventional train/val/test splits for each benchmark. Besides, we set the aggregation weight of the original image as 0.5, and additional images as 0.5 altogether, so as to emphasize the original images. For diffusion noise addition, we set the noise level to 0.7. Details are available in Appendix Section §I.

**Evaluation Protocol.** We evaluate three standard backbones used in FSL: a shallow 4-layer convolutional neural network (ConvNet), a 12-layer residual network (Res12) (He et al., 2016), a small vision transformer backbone (ViTSmall) (Dosovitskiy et al., 2021) and a tiny swin transformer backbone (SwinTiny) (Liu et al., 2021). Note that the encoders ViTSmall and SwinTiny are pretrained on the ImageNet-1k Russakovsky et al. (2015) dataset, while the Res12 and ConvNet

encoders on trained on the train-split of each FSL dataset. Classification is performed with a non-parametric Euclidean-distance/cosine-similarity prototype classifier in the episodic setting. These choices match common FSL practice, including ProtoNet and FEAT-style set-to-set variants that operate on support/query embeddings, ensuring comparability with prior work and our reproduced baselines (Snell et al., 2017; Ye et al., 2020). We evaluate off-the-shelf trained ProtoNet and FEAT models. We adopt Euclidean-distance-based classifier for miniImagenet and tieredImagenet, and cosine-similarity-based classifier for Animals and CUB. Our reproduced baselines closely match prior reports evaluation set-up (e.g., DeepEMD (Zhang et al., 2020), Meta-Baseline (Chen et al., 2021), MetaOptNet (Lee et al., 2019) on 5-way-1/5-shot inductive benchmarks, all with the Res12 backbone. Among the baselines, SLA-AG Lee et al. (2020) involves self-supervised label augmentation, and Meta-MaxUp Ni et al. (2020) involves training data augmentation, both being ensemble-based methods.). We sample 15,000 5-way-1/5-shot queries and report mean accuracy with 95% confidence intervals across episodes. This follows the standard protocol used in related FSL work (Ye et al., 2020; Snell et al., 2017).

| Method (Res12) | 5-Way-1-Shot (%) | | 5-Way-5-Shot (%) | |
| --- | --- | --- | --- | --- |
| | miniImagenet | tieredImagenet | miniImagenet | tieredImagenet |
| DeepEMD Zhang et al. (2020) | $65.91 \pm 0.82$ | $71.16 \pm 0.87$ | $82.41 \pm 0.56$ | **86.03** $\pm 0.58$ |
| Meta-MaxUp Ni et al. (2020) | $62.81 \pm 0.34$ | - | $79.38 \pm 0.24$ | - |
| Meta-Baseline Chen et al. (2021) | $63.17 \pm 0.23$ | $68.62 \pm 0.27$ | $79.26 \pm 0.17$ | $83.74 \pm 0.18$ |
| MetaOptNet Lee et al. (2019) | $62.64 \pm 0.61$ | $65.99 \pm 0.72$ | $78.63 \pm 0.46$ | $81.56 \pm 0.53$ |
| SLA-AG Lee et al. (2020) | $62.93 \pm 0.63$ | — | $79.63 \pm 0.47$ | — |
| ProtoNet + TRAML Li et al. (2020) | $60.31 \pm 0.48$ | — | $77.94 \pm 0.57$ | — |
| ConstellationNet Xu et al. (2021) | $64.89 \pm 0.23$ | — | $79.95 \pm 0.17$ | — |
| Classifier-Baseline Chen et al. (2021) | $58.91 \pm 0.23$ | $68.07 \pm 0.26$ | $77.76 \pm 0.17$ | $83.74 \pm 0.18$ |
| DFR Cheng et al. (2023) | **67.74** $\pm 0.86$ | **71.31** $\pm 0.93$ | **82.49** $\pm 0.57$ | $85.12 \pm 0.64$ |
| ProtoNet-Res12 Snell et al. (2017) | $62.39 \pm 0.21$ | $68.23 \pm 0.23$ | $80.53 \pm 0.14$ | $84.03 \pm 0.16$ |
| ProtoNet-Res12 (re-impl.) | $60.01 \pm 0.65$ | $65.28 \pm 0.32$ | $75.34 \pm 0.49$ | $81.13 \pm 0.29$ |
| ProtoNet-Res12+1S-DAug-1 (Ours) | $62.90 \pm 0.66$ *(+2.89%↑)* | $69.06 \pm 0.32$ *(+3.78%↑)* | $78.89 \pm 0.48$ *(+3.55%↑)* | $83.86 \pm 0.27$ *(+2.73%↑)* |
| ProtoNet-Res12+1S-DAug-2 (Ours) | $64.61 \pm 0.66$ *(+4.60%↑)* | $70.32 \pm 0.32$ *(+5.04%↑)* | - | - |
| ProtoNet-Res12+1S-DAug-3 (Ours) | $64.94 \pm 0.66$ *(+4.93%↑)* | - | - | - |
| FEAT-Res12 Ye et al. (2020) | $66.78 \pm 0.20$ | $70.80 \pm 0.23$ | $82.05 \pm 0.14$ | $84.79 \pm 0.16$ |
| FEAT-Res12 (re-impl.) | $63.31 \pm 0.65$ | $68.28 \pm 0.28$ | $77.90 \pm 0.48$ | $82.21 \pm 0.28$ |
| FEAT-Res12+1S-DAug-1 (Ours) | $67.08 \pm 0.65$ *(+3.77%↑)* | $71.85 \pm 0.28$ *(+3.57%↑)* | $81.96 \pm 0.44$ *(+4.06%↑)* | $84.82 \pm 0.26$ *(+2.61%↑)* |
| FEAT-Res12+1S-DAug-2 (Ours) | $69.04 \pm 0.65$ *(+5.73%↑)* | **73.18** $\pm 0.27$ *(+4.90%↑)* | $82.62 \pm 0.45$ *(+4.72%↑)* | **85.55** $\pm 0.25$ *(+3.34%↑)* |
| FEAT-Res12+1S-DAug-3 (Ours) | **69.25** $\pm 0.65$ *(+5.94%↑)* | - | **83.38** $\pm 0.41$ *(+5.48%↑)* | - |

Table 1: Inductive 5-way-1-shot and 5-way-5-shot accuracy (%) on miniImagenet and tieredImagenet with Res12 backbones. Dashes denote unavailable or less important results not reported. The best results of ours and other FSL methods are both highlighted in bold. Our method transforms the weaker models to become stronger than most of the other Res12 baselines; we can likely achieve even better performance using stronger base models.

| Method (Res12/ConvNet) | Animals | CUB |
| --- | --- | --- |
| ProtoNet | $73.20 \pm 0.63$ | $46.38 \pm 0.22$ |
| ProtoNet+1S-DAug-2 (Ours) | $75.20 \pm 0.65$ *(+2.00%↑)* | $55.50 \pm 0.24$ *(+9.12%↑)* |
| FEAT | $79.37 \pm 0.59$ | $51.10 \pm 0.24$ |
| FEAT+1S-DAug-2 (Ours) | **80.66** $\pm 0.62$ *(+1.23%↑)* | **61.55** $\pm 0.25$ *(+10.45%↑)* |

Table 2: Inductive 5-way-1-shot accuracy (%) on Animals with Res12 backbones and CUB with ConvNet backbones.

| Dataset | Method | ViTSmall | SwinTiny |
| --- | --- | --- | --- |
| MiniImagenet | ProtoNet Snell et al. (2017) | 71.86 | 67.32 |
| | ProtoNet+1S-DAug-1 (Ours) | 80.42 *(+8.56%↑)* | 75.12 *(+7.80%↑)* |
| | ProtoNet+1S-DAug-2 (Ours) | 82.76 *(+10.90%↑)* | 77.82 *(+10.50%↑)* |
| | ProtoNet+1S-DAug-3 (Ours) | 83.66 *(+11.80%↑)* | 78.92 *(+11.60%↑)* |
| CUB | ProtoNet Snell et al. (2017) | 71.90 | 69.78 |
| | ProtoNet+1S-DAug-1 (Ours) | 75.72 *(+3.82%↑)* | 71.76 *(+1.98%↑)* |

Table 3: 5-way-1-shot accuracy (%) on miniImagenet/CUB with ViTSmall/SwinTiny backbones.

**Main Results.** Table 1/2/3 summarizes 5-way-1-shot and 5-way-5-shot results with Res12/ConvNet/ViTSmall/SwinTiny backbones, and our method with 1/2/3 additional augmentations are denoted as 1S-DAug-1/2/3 respectively. Note that we directly adopt 5-way-1-shot FSL models for the 5-way-5-shot evaluation, and Table 2 contains 5-way-1-shot results on CUB and Animals. As reported, test-time 1S-DAug consistently improves over the corresponding non-augmented baselines and over prior strong Res12 methods reported under the same backbone (e.g., on miniImagenet, FEAT improves from 63.31% to 69.25%, a maximum absolute gain of +5.94 percentage points, which achieves the highest among all the reported FSL works with Res12 backbones). The gains persist across both datasets (e.g., miniImagenet improves by at least +2.89%, tieredImagenet by +3.88%, CUB by +9.12%, and Animals by +1.23% on the 5-way-1-shot benchmarks), implying a high probability for: (i) the image-conditioned diffusion step preserves class-defining content sufficiently, and (ii) the shape tweak with noising creates diversity without compromising faithfulness.

## 6.2 ABLATION STUDIES

We conduct ablation studies for our method, and more analyses, including hyperparameter tuning (Section §B), efficiency studies (Section §A) and limitation (Section §C) are available in Appendix.

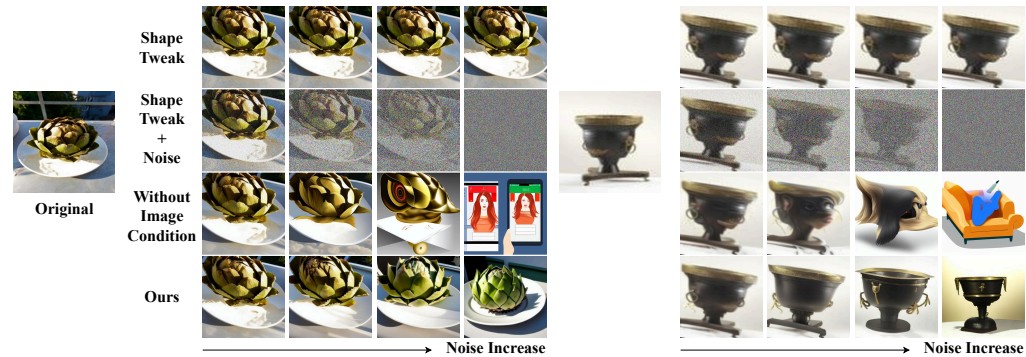

Figure 2: Effect of noise and conditioning. Qualitative ablation on a single input across increasing noise levels. Shape-only edits yield limited diversity; adding noise increases diversity but may reduce fidelity without conditioning. Attention-conditioned diffusion preserves class-defining content while enabling controlled pose/appearance changes; excessive noise without the image condition degrades faithfulness.

**Qualitative Analysis.** Figure 2 illustrates the effect of removing image conditioning, removing shape tweaks, and sweeping the noise level. With very small noise, changes are minimal; with very large noise and no image conditioning, generations may drift toward off-class content; removing shape tweaks reduces diversity and visible pose/layout variations. Additional visualizations are provided in the Appendix Section §K.

**Quantitative Analysis.** We further dissect the contribution of each component, including aggregation weight adjustment, image conditioning, noise magnitude, shape tweaking and diffusion generation using FEAT with a Res12 backbone on miniImagenet, in a controlled 5-way-1-shot setting with one augmented query and one augmented support per episode (Table 4). We first notice that reducing the emphasis on original samples via less aggregation weight would downgrade model accuracy slightly. This is expected, since the original samples are authentic images with the best quality. Besides, removing the image conditioning downgrades the performance severely, which mirrors our qualitative studies in Figure 2. When a small noise level ($\eta$=0.20) is applied with shape tweaking and conditioning, diversity gain is limited, and distortion caused by shape tweaking may also stay and backfire, yielding 63% accuracy. Increasing the noise strength to a moderate level ($\eta$=0.70) improves coverage while preserving class faithfulness, pushing performance to 67.1%, the best among diffusion-based rows. Pushing noise to the extreme ($\eta$=1.0) still delivers reasonable performance (67.0%) when conditioning is enabled. Such a performance is enabled by the

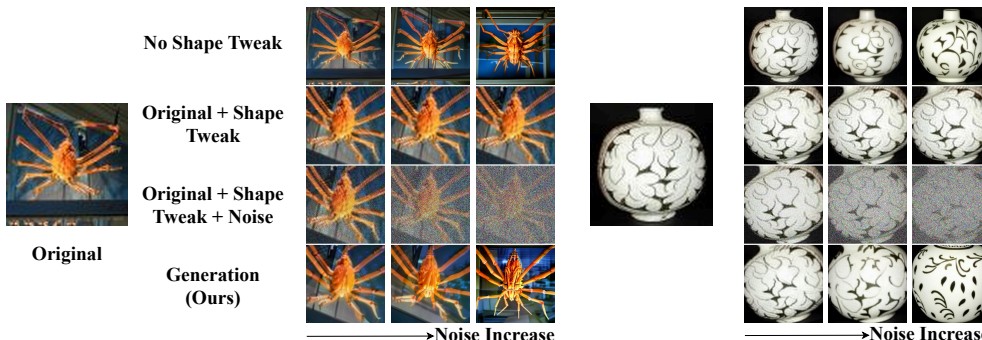

Figure 3: Effect of noise and shape tweak. Comparison across three settings: no shape tweak, shape tweak only, and shape + noise + attention-conditioned diffusion (ours). Increasing noise and including shape tweak expand diversity, and our full setting provides the best balance for both diversity and faithfulness.

greater diversity between the generation output and the original input, but the originality faithfulness is jeopardized, especially when confronted with rare object types. Therefore, generation from full noise should be discouraged. Besides, ablating shape tweaks reduces accuracy to 66.1%, confirming that geometric variation helps cover different pose/layout. Substituting true extra images of the same class ("Real/Oracle") provides an upper bound of 78.0%, showing the headroom available with more independent samples. Meanwhile, traditional geometric test-time edits (rotations, affine warps, color jitter) only reach 57.89%, supporting the observation that such transforms add little new information and may distort original images.

| Same Noise Level | Shape Tweak | Generation Techniques | Additional Adjustment | 5-Way-1-Shot Accuracy |
|---|---|---|---|---|
| ✓(0.70) | ✓ | ✓ | ✓(0.3 original image weight) | $66.79 \pm 0.63$ |
| ✓(0.70) | ✓ | ✓ | ✓(remove image conditioning) | $53.94 \pm 0.62$ |
| ×(0.20) | ✓ | ✓ | × | $63.45 \pm 0.64$ |
| ✓(0.70) | ✓ | ✓ | × | $67.08 \pm 0.62$ |
| ×(1.00) | - | ✓ | × | $67.01 \pm 0.68$ |
| ✓(0.70) | × | ✓ | × | $66.12 \pm 0.66$ |
| - | ✓(Real/Oracle) | × | × | $77.99 \pm 0.62$ |
| - | ✓(Traditional) | × | × | $57.89 \pm 0.67$ |

Table 4: Ablation of aggregation weight adjustment, shape tweak, noise level, and diffusion conditioning. 'Traditional' uses standard geometric edits; 'Real' substitutes actual additional images of the same object type. Our full setting (shape + controlled noise + attention-conditioned diffusion) outperforms classical test-time augmentation and approaches the oracle (true image) upper bound.

## 7 CONCLUSIONS AND FUTURE WORK

To conclude, we presented *1S-DAug*, a one-shot, test-time generative augmentation operator that synthesizes diverse yet faithful variants from a single image. By combining geometric perturbations with controlled noising and attention-conditioned denoising, the method maintains class-defining content while enhancing data diversity. As a plugin into standard FSL models, 1S-DAug delivers consistent non-trivial accuracy gains under the 5-way-1/5-shot protocol across datasets, without FSL training, fine-tuning, or access to specific model parameters. This model-agnostic, data-side design makes the approach practical for modern deployments where models are large, fixed, or restricted. While our focus is test-time augmentation for FSL, the operator is also applicable to other downstream tasks like training-time augmentation and controllable image editing. Future work will explore inference speedups, more backbones and constraining diffusion-based image generator data to within the FSL training splits. Overall, we believe 1S-DAug offers a useful building block for data-centric robustness with high potential in real-world low-label scenarios.

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

**Ethics Statement.** We affirm adherence to the ICLR Code of Ethics. Our study uses only standard, publicly available benchmarks (miniImagenet, tieredImagenet, CUB, and an animal-face dataset referenced in prior work) under their respective licenses. No new data were collected, no personally identifiable or sensitive information is involved, and no human-subject interventions were conducted; institutional review board approval was therefore not required. The proposed method is a test-time, data-side augmentation wrapper intended to improve recognition robustness in few-shot settings; it does not require access to underlying model parameters. Potential risks include misuse of generative models to synthesize misleading content and amplification of dataset biases. To mitigate these risks, we (i) confine generation to class-faithful, small perturbations of inputs, (ii) evaluate on public benchmarks with well-documented splits, and (iii) plan to release code with conservative defaults and documentation describing appropriate use and limitations. We disclose no conflicts of interest or external sponsorship affecting the work.

**Reproducibility Statement.** We take reproducibility seriously. The paper specifies the method mathematically (Section § 4), the experimental set-up (encoders, classifier, datasets, and episodic protocol), and all evaluation details (Section § 6); ablations and qualitative analyses are provided to validate design choices. The Appendix Section §I details data preprocessing (including upscaling and shape-tweak parameters), hyperparameters (noise level, conditioning strength, denoising steps), and the exact episodic sampling procedure (5-way-1-shot, 15,000 queries, 95% confidence intervals). Upon paper acceptance, we will release a repository containing: training scripts for ProtoNet/FEAT under the stated backbones, inference scripts for our augmentation, configuration files for table, deterministic seeds, and instructions to download datasets and reproduce numbers end-to-end on a single GPU (the hardware we report). Where we use pretrained weights or models, we provide pointers or scripts to obtain them. Together, these materials enable exact regeneration of all reported tables and figures.

## A    EFFICIENCY

|  | **0.25 Noise** | **0.50 Noise** | **0.75 Noise** | **1.00 Noise** |
|---|---|---|---|---|
| **Generation Time (s)** $\downarrow$ | 0.41 | 0.68 | 0.92 | 1.42 |

Table 5: Per-image generation time across noise levels. Higher noise entails more denoising compute. Measured on a single GPU; see experimental setup for hardware details.

We record the wall-clock running time for our inference script, and the results are reported in Table 5. All experiments are run on a single NVIDIA RTX A5000 GPU. As expected, runtime scales with the noising level. Higher noising (and more denoising steps) produces larger edits and requires longer inference, whereas lower noising is faster. As we start from the noisy image halfway, the inference cost is generally lower than that of standard diffusion process starting from pure noise.

## B    HOW MUCH AND WHERE TO AUGMENT

We ablate support-only, query-only, and joint support+query augmentation under ProtoNet (Res12) on miniImagenet. See our ablation table (Table 4) for the full grid. Accuracy improves monotonically as we add a number of augmented copies to *both* support and queries (e.g., from 60.01% with no augmentation to 64.94% with +3/3, yielding a +4.93 absolute gain). Adding only support copies while leaving queries un-augmented can underperform due to distribution mismatch between prototype construction and query embeddings (e.g., with +3/0 accuracy is 61.82%, well below the 64.55% achieved when queries are matched with +3/1). This suggests that matched augmentation on both sides yields the largest benefit.

## C    LIMITATIONS

Our approach has two main limitations. First, some 1S-DAug evaluation involves pretrained components, including ViTSmall Dosovitskiy et al. (2021) and SwinTiny Liu et al. (2021) encoders pretrained on ImageNet-1k Russakovsky et al. (2015) and a Stable-Diffusion-v1.5 generator Rombach

|  | | Query | | | |
|---|---|---|---|---|---|
|  | | **+0** | **+1** | **+2** | **+3** |
| **Support** | +0 | $60.01 \pm 0.65$ | $60.24 \pm 0.69$ | $60.09 \pm 0.70$ | $60.31 \pm 0.70$ |
|  | +1 | $60.20 \pm 0.66$ | $62.90 \pm 0.66$ | $63.05 \pm 0.67$ | $63.16 \pm 0.67$ |
|  | +2 | $61.80 \pm 0.64$ | $64.53 \pm 0.66$ | $64.61 \pm 0.66$ | $64.87 \pm 0.66$ |
|  | +3 | $61.82 \pm 0.64$ | $64.55 \pm 0.65$ | $64.72 \pm 0.66$ | $64.94 \pm 0.66$ |

Table 6: Inductive 5-way-1-shot accuracy (mean ± 95% CI) as a function of the number of augmented copies for supports (rows) and queries (columns).

et al. (2022a). Therefore, we cannot fully rule out potential data leakage; future work will explore strict constraint within the few-shot training splits. Second, the diffusion-based augmentation introduces inference overhead compared to running the backbone alone or using classical geometric augmentations. Reducing this computational cost (e.g., via lighter generative backbones or faster denoising schedules) is an important direction for future work.

## D  MORE RELATED WORK

**Few-shot Learning.**  FSL methods commonly fall into metric-, model-, and augmentation-based families. *Metric-based* methods learn an embedding where queries are classified by proximity to supports or class prototypes, including Matching Networks (Vinyals et al., 2016), Prototypical Networks (Snell et al., 2017), Relation Networks (Sung et al., 2018), and episodic feature adaptation such as FEAT (Ye et al., 2020). Strong baselines refine this recipe with improved training protocols and heads, e.g., Baseline++ (Chen et al., 2019), Meta-Baseline (Chen et al., 2021), MetaOptNet (Lee et al., 2019), and transductive inference methods such as TPN (Liu et al., 2019b), LaplacianShot (Ziko et al., 2020), and TIM (Iscen et al., 2020). *Model-based* approaches emphasize rapid parameter adaptation from few examples, e.g., gradient-based meta-learning with MAML (Finn et al., 2017), Meta-SGD (Li et al., 2017), Reptile (Nichol et al., 2018), and ANIL (Raghu et al., 2020). *Augmentation-based* approaches increase training diversity via feature or image synthesis—e.g., feature hallucination (Hariharan & Girshick, 2017b) and delta-based example synthesizers (Schwartz et al., 2018). These are primarily *training-time* techniques that rely on base-class supervision; by contrast, few-shot *test-time* augmentation must produce high-quality, class-faithful variants for unseen classes without retraining or labels.

Test-time generative augmentation for FSL remains limited. Bai et al. (2025) uses an adversarial image-to-image translator to combine the geometric "shape" of one image with the class-defining "style" of another (i.e., FUNIT (Liu et al., 2019a)) for inference-time augmentation. While a useful proof of concept, the dataset scope is narrow and failure arises on more complex, diverse categories, reflecting the difficulty of preserving content under large structural gaps.

**Diffusion Models.**  Early diffusion models established iterative denoising as a competitive generative paradigm (Sohl-Dickstein et al., 2015; Ho et al., 2020), with subsequent improvements to training and sampling (Nichol & Dhariwal, 2021). Latent-space diffusion amortizes computation via a learned autoencoder, enabling high-resolution synthesis (Rombach et al., 2022b). Attention-based conditioning adapters inject external signals into cross-attention without retraining the denoiser, supporting controllable editing and image-conditioned generation, including image-prompt adapters (Ye et al., 2023), general adapters (Mou et al., 2023), and control modules such as ControlNet (Zhang et al., 2023). These advances in stability, controllability, and fidelity make diffusion well-suited for few-shot test-time augmentation. Editing-by-denoising constructs variants by adding controlled noise to a source image and running the reverse process with conditioning (Meng et al., 2022). However, in this setup, too little noise yields small changes, and too much sacrifices faithfulness (e.g., changes object type), which is not suitable for data augmentation.

**GAN/Diffusion-based Data Augmentation.**  Adversarial generators have long been used for data expansion and translation. Few-shot translation frameworks (e.g., FUNIT (Liu et al., 2019a) and derivatives), unpaired mappers (CycleGAN (Zhu et al., 2017)), and class-conditional generators (StyleGAN families (Karras et al., 2019)) can expand training sets but face limitations for FSL test-

---

**Algorithm 1** 1S-DAug (Single image $x$)

---

**Require:** image $x$; steps $T$; schedule $(\beta_t)$; user noise $\eta \in [0, 1]$; conditioning weight $\lambda_{\text{img}}$; number of variants $K$; optional text prior $p$
 1: **Geometric seed:** sample a shape tweak $T_\psi$ and set $x_{\text{geom}} \leftarrow T_\psi(x)$
 2: **Noising entry:** compute $t_0$ from $\eta$ via equation equation 2; draw $x_{t_0} \sim q(\cdot \mid x_{\text{geom}})$ using equation equation 1
 3: **Working state:** set $z_{t_0} \leftarrow x_{t_0}$ (pixel space) or $z_{t_0} \leftarrow \text{Enc}(x_{t_0})$ (latent variant)
 4: **for** $k = 1$ to $K$ **do**
 5:     **for** $t = t_0, t_0 - 1, \ldots, 1$ **do**
 6:         Form $K_t, V_t$ by equation equation 7 using $x$ (and $p$ if used); set $Q_t = W_Q z_t$ and compute $c_t \leftarrow A_t(Q_t, K_t, V_t)$
 7:         **Reverse step:** update $z_{t-1} \leftarrow \mu_\varphi(z_t, t, c_t) + \sigma_t \epsilon$ via equation equation 3, with $\epsilon \sim \mathcal{N}(0, I)$
 8:     **end for**
 9:     **Decode:** $\tilde{x}^{(k)} \leftarrow \text{Dec}(z_0)$                   ▷ identity if denoising in pixel space
10: **end for**
11: **return** $\{\tilde{x}^{(k)}\}_{k=1}^K$

---

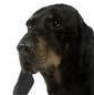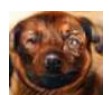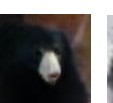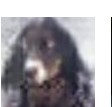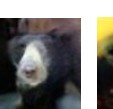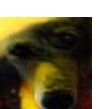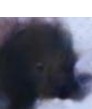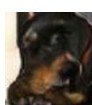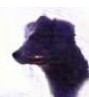

**Intended Class**          **Failure Generation (GAN)**

Figure 4: Failure modes of GAN-based image-to-image translation. Examples where image-to-image GAN translation fails to preserve the intended class. Rows contain target class and failed GAN outputs with typical artifacts.

time augmentation, including training stability, mode coverage, and faithfulness for *unseen* classes without supervision. For diffusion-based generators, there have been a few recent works focusing on using the diffusion-based synthetic images for downstream tasks other than few-shot learning. However, these works rely on fine-tuning with text prompts or a handful of extra target-class samples, not suitable for test-time augmentation (He et al., 2023; Benigmim et al., 2023). In contrast, we do not rely on any label or additional target-class samples, and the strict set-up fulfills the requirement for challenging downstream tasks like FSL test-time augmentation.

## E   ALGORITHM SUMMARY

Algorithmic summary of 1SDAug is available as Algorithm 1.

## F   GAN FAILURE CASE ILLUSTRATION

Figure 4 illustrates the failure cases of GAN-based image-to-image translation models (Liu et al., 2019a).

## G   PROOF OF PROPOSITION RISK DECOMPOSITION

Recall that $R(g) = \frac{1}{4} \mathbb{E}\big[(g(x) - y)^2\big]$ and $f, f_A : \mathcal{X} \to \{-1, 1\}$, $\tilde{f} = \frac{1}{2}(f + f_A)$. Since $f^2(x) = f_A^2(x) = y^2 = 1$,

$$R(f) = \frac{1}{4} \mathbb{E}\big[(f - y)^2\big] = \frac{1}{4} \mathbb{E}[f^2 - 2fy + y^2] = \frac{1}{2} - \frac{1}{2} \mathbb{E}[f(x)y]. \tag{15}$$

For $\tilde{f}$,

$$R(\tilde{f}) = \frac{1}{4}\,\mathbb{E}\Big[\Big(\frac{f+f_A}{2}-y\Big)^2\Big] = \frac{1}{16}\,\mathbb{E}\big[(f+f_A)^2\big] - \frac{1}{4}\,\mathbb{E}\big[(f+f_A)y\big] + \frac{1}{4}\,\mathbb{E}[y^2]. \qquad (16)$$

Expanding $(f+f_A)^2 = f^2 + 2ff_A + f_A^2$ and using $f^2 = f_A^2 = y^2 = 1$,

$$R(\tilde{f}) = \frac{1}{16}\,\mathbb{E}[2+2ff_A] - \frac{1}{4}\,\mathbb{E}[fy+f_Ay] + \frac{1}{4} = \frac{1}{8}\big(1+\mathbb{E}[f(x)f_A(x)]\big) - \frac{1}{4}\,\mathbb{E}[f(x)y] - \frac{1}{4}\,\mathbb{E}[f_A(x)y] + \frac{1}{4}.$$
$$(17)$$

Subtracting $R(f)$ gives

$$R(\tilde{f}) - R(f) = \frac{1}{4}\big(\mathbb{E}[f(x)y] - \mathbb{E}[f_A(x)y]\big) + \frac{1}{8}\big(\mathbb{E}[f(x)f_A(x)] - 1\big). \qquad (18)$$

# H  ADDITIONAL THEORETICAL RESULTS

Throughout this appendix we use the pairwise reduction of Section 5.1. Each input is a query–prototype pair $x = (q,p)$ with label $y \in \{-1,1\}$. For a fixed encoder $\Phi_\theta$ and prototype $p$ we define the difference feature $\Omega_\theta(x) := \Phi_\theta(q) - p$ and the Euclidean score $g_\theta(x) := -\|\Omega_\theta(x)\|_2^2$. The 0–1 pairwise risk of $\theta$ is

$$R_{\mathrm{cls}}(\theta) := \mathbb{P}\big(y\,g_\theta(x) \le 0\big),$$

which coincides with the classification risk used in Section 5.3.

Where the arguments apply to generic real-valued predictors, we write $g$ for a function in a class $\mathcal{G}$ and specialize to the encoder score class $\mathcal{G} = \{x \mapsto g_\theta(x) : \theta \in \Theta\}$ at the end.

## H.1  EPISODIC EUCLIDEAN MODEL AND TEST-TIME AUGMENTATION (DETAILS)

An $N$-way $K$-shot episode has support set $S = \{(s_{c,k},c) : c = 1,\ldots,N,\ k = 1,\ldots,K\}$ and query set $Q = \{(q_j,y_j)\}_{j=1}^{n_q}$ with $y_j \in \{1,\ldots,N\}$. A single encoder $\Phi_\theta : \mathcal{X} \to \mathbb{R}^d$ produces features $z_{c,k} := \Phi_\theta(s_{c,k})$ and $z_q := \Phi_\theta(q)$. Class prototypes and the Euclidean classifier are

$$p_c := \frac{1}{K}\sum_{k=1}^{K} z_{c,k}, \qquad \hat{y}(q) := \arg\min_c \|\Phi_\theta(q) - p_c\|_2^2,$$

where $p_c$ corresponds to $p_c$ in Section 4. For analysis we reduce episodes to binary query–prototype pairs: $x = (q,p)$ with label $y \in \{-1,1\}$, where $y = +1$ if $p$ is the prototype of the true class of $q$ and $y = -1$ otherwise. We work with the difference feature $\Omega_\theta(x) := \Phi_\theta(q) - p$ and Euclidean score $g_\theta(x) := -\|\Omega_\theta(x)\|_2^2$, and assume a uniform pre-augmentation radius bound

$$\|\Omega_\theta(x)\|_2 \le r_0 \qquad \text{for all } x \text{ and } \theta,$$

as in Section 5.1.

At test time, 1S-DAug uses support-side feature averaging and query-side logit averaging as described in Section 4, yielding (possibly aggregated) prototypes $p_c$. For the theory we only need query-side notation. Let $A_k(q)$ be the $k$-th augmented view of a query $q$ ($k = 0,\ldots,K_a$), $z^{(k)}(q) := \Phi_\theta(A_k(q))$, and $(\alpha_k)_{k=0}^{K_a}$ convex weights with $\sum_k \alpha_k = 1$. We define the averaged query embedding

$$\bar{z}_{\mathrm{qry}}(q) := \sum_{k=0}^{K_a} \alpha_k z^{(k)}(q),$$

the aggregated difference feature $\bar{\Omega}_\theta(x) := \bar{z}_{\mathrm{qry}}(q) - p$, and the aggregated score $\bar{g}_\theta(x) := -\|\bar{\Omega}_\theta(x)\|_2^2$, where $p$ denotes a (possibly aggregated) prototype. For squared Euclidean scores, logit averaging over per-view scores $g_\theta^{(k)}(q,p) = -\|z^{(k)}(q) - p\|_2^2$ is equivalent (up to a class-independent constant) to using $\bar{z}_{\mathrm{qry}}(q)$; see Appendix H.3. We therefore express all bounds in terms of $\bar{\Omega}_\theta(x)$ and $\bar{g}_\theta(x)$.

## H.2 Margin-based bound and Rademacher complexity

For completeness we recall the margin loss and Rademacher complexity used in Section 5. For a score function $g_\theta$ and a pair $(x, y) \sim D$ with $y \in \{-1, 1\}$, define the signed margin $u_\theta(x) := y\, g_\theta(x)$. Fix a margin parameter $\rho > 0$ and the piecewise-linear margin loss

$$\tau_\rho(t) := \begin{cases} 1, & t \le 0, \\ 1 - t/\rho, & 0 < t < \rho, \\ 0, & t \ge \rho. \end{cases}$$

The (population) margin risk and empirical margin risk on a sample $S = \{(x_i, y_i)\}_{i=1}^m$ are

$$R_\rho(\theta) := \mathbb{E}[\tau_\rho(u_\theta(x))], \qquad \widehat{R}_{S,\rho}(\theta) := \frac{1}{m} \sum_{i=1}^m \tau_\rho(y_i g_\theta(x_i)).$$

Let the empirical 0–1 pairwise risk on $S$ be

$$\widehat{R}_{\mathrm{cls}, S}(\theta) := \frac{1}{m} \sum_{i=1}^m \mathbf{1}\{y_i g_\theta(x_i) \le 0\}.$$

Since $\mathbf{1}\{t \le 0\} \le \tau_\rho(t)$, we have

$$R_{\mathrm{cls}}(\theta) \le R_\rho(\theta) \quad \text{and} \quad \widehat{R}_{\mathrm{cls}, S}(\theta) \le \widehat{R}_{S,\rho}(\theta).$$

Let $\mathcal{G} := \{g_\theta : \theta \in \Theta\}$ be the encoder score class. Its empirical Rademacher complexity is

$$\widehat{\mathfrak{R}}_S(\mathcal{G}) := \mathbb{E}_\sigma \Big[ \sup_{g \in \mathcal{G}} \frac{1}{m} \sum_{i=1}^m \sigma_i g(x_i) \Big],$$

where $\sigma_i$ are independent Rademacher variables. A standard contraction argument with $\tau_\rho$ yields Theorem 1 in the main text:

**Theorem 2** (Restatement of Theorem 1). *For any $\rho > 0$ and $\delta > 0$, with probability at least $1 - \delta$ over $S \sim D^m$, every $\theta$ satisfies*

$$R_{\mathrm{cls}}(\theta) \le \widehat{R}_{S,\rho}(\theta) + \frac{2}{\rho} \widehat{\mathfrak{R}}_S(\mathcal{G}) + \sqrt{\frac{\log(1/\delta)}{2m}}.$$

**Lemma 2** (Rademacher complexity of the encoder score class). *Suppose each encoder $\Phi_\theta$ is realised by a feedforward network with 1-Lipschitz nonlinearities and layer spectral norms $\|W_\ell\|_2 \le s_\ell$ such that $\prod_{\ell=1}^L s_\ell \le L_{\mathrm{enc}}$, and that $\|\Omega_\theta(x)\|_2 \le r$ for all $x$ and $\theta$. Then there exists a constant $C_{\mathrm{enc}} > 0$ (depending only on the architecture) such that, for any sample $S$,*

$$\widehat{\mathfrak{R}}_S(\mathcal{G}) \le C_{\mathrm{enc}} L_{\mathrm{enc}} \frac{r}{\sqrt{m}}.$$

*In particular, in the setting of Section 5.3, one may take $r = r_0$ before augmentation and $r = \hat{r}$ after augmentation.*

*Proof sketch.* By assumption, each encoder $\Phi_\theta$ is realised by a feedforward network with 1-Lipschitz nonlinearities and layer spectral norms $\|W_\ell\|_2 \le s_\ell$ satisfying $\prod_{\ell=1}^L s_\ell \le L_{\mathrm{enc}}$. Standard results on spectral-norm control of deep networks (e.g. via composing linear maps and 1-Lipschitz activations) imply that $\Phi_\theta$ is $L_{\mathrm{enc}}$-Lipschitz with respect to the input $\ell_2$-norm, i.e.

$$\|\Phi_\theta(x) - \Phi_\theta(x')\|_2 \le L_{\mathrm{enc}} \|x - x'\|_2 \qquad \text{for all } x, x'.$$

Since $p$ does not depend on $q$ for a fixed pair $x = (q, p)$, the difference feature $\Omega_\theta(x) = \Phi_\theta(q) - p$ is also $L_{\mathrm{enc}}$-Lipschitz in $q$. On the domain where $\|\Omega_\theta(x)\|_2 \le r$ for all $x$ and $\theta$, the score $g_\theta(x) = -\|\Omega_\theta(x)\|_2^2$ is $2r L_{\mathrm{enc}}$-Lipschitz in $x$: the gradient of $g_\theta$ with respect to $\Omega_\theta$ has norm $2\|\Omega_\theta(x)\|_2 \le 2r$, and $\Omega_\theta$ itself is $L_{\mathrm{enc}}$-Lipschitz.

Let $\mathcal{G} = \{g_\theta : \theta \in \Theta\}$ be the corresponding score class. It is therefore contained in a class of Lipschitz real-valued functions whose Lipschitz constant is bounded by $2rL_{\text{enc}}$. Standard covering-number/Rademacher arguments for such Lipschitz function classes (see, e.g., generic chaining bounds for $L$-Lipschitz functions on an $\ell_2$-ball) yield that there exists a constant $C > 0$, depending only on the input dimension and hence only on the architecture, such that for any sample $S$ of size $m$,

$$\widehat{\mathfrak{R}}_S(\mathcal{G}) \leq C\left(2rL_{\text{enc}}\right)\frac{1}{\sqrt{m}}.$$

Setting $C_{\text{enc}} := 2C$ gives

$$\widehat{\mathfrak{R}}_S(\mathcal{G}) \leq C_{\text{enc}} L_{\text{enc}} \frac{r}{\sqrt{m}},$$

which is the claimed bound. $\qquad\square$

### H.3 EQUIVALENCE OF LOGIT AVERAGING AND FEATURE AVERAGING

For completeness we record the standard equivalence between logit averaging and feature averaging for squared Euclidean scores.

Fix a query $q$ and a prototype $p$, and let

$$z^{(k)}(q) := \Phi_\theta\left(A_k(q)\right)$$

denote the encoded query under augmentation $k$, with convex weights $\alpha_k \geq 0$, $\sum_k \alpha_k = 1$. The per-view scores are

$$g_\theta^{(k)}(q,p) := -\left\|z^{(k)}(q) - p\right\|_2^2, \qquad \tilde{g}_\theta(q,p) := \sum_k \alpha_k\, g_\theta^{(k)}(q,p).$$

Define the averaged query feature

$$\bar{z}(q) := \sum_k \alpha_k\, z^{(k)}(q), \qquad g_\theta^{\text{avg}}(q,p) := -\left\|\bar{z}(q) - p\right\|_2^2.$$

Then

$$\begin{aligned}
\tilde{g}_\theta(q,p) &= -\sum_k \alpha_k\left\|z^{(k)}(q) - p\right\|_2^2 \\
&= -\sum_k \alpha_k\left(\|z^{(k)}(q)\|_2^2 - 2\langle z^{(k)}(q), p\rangle + \|p\|_2^2\right) \\
&= -\sum_k \alpha_k\|z^{(k)}(q)\|_2^2 + 2\left\langle \sum_k \alpha_k z^{(k)}(q), p\right\rangle - \|p\|_2^2 \\
&= -\sum_k \alpha_k\|z^{(k)}(q)\|_2^2 - \|\bar{z}(q)\|_2^2 + \|\bar{z}(q)\|_2^2 + 2\langle\bar{z}(q), p\rangle - \|p\|_2^2 \\
&= -\left\|\bar{z}(q) - p\right\|_2^2 - \sum_k \alpha_k\|z^{(k)}(q)\|_2^2 + \|\bar{z}(q)\|_2^2.
\end{aligned}$$

The last two terms depend on $q$ and the set of views $\{z^{(k)}(q)\}$ but not on $p$. Hence, for fixed $q$, comparing classes by $\tilde{g}_\theta(q, p_c)$ is equivalent to comparing them by

$$g_\theta^{\text{avg}}(q, p_c) = -\left\|\bar{z}(q) - p_c\right\|_2^2.$$

In other words, query-side logit averaging with squared Euclidean scores induces exactly the same class ranking as nearest-prototype classification in the feature space of the averaged query embedding $\bar{z}(q)$.

This justifies working, for analysis, with the aggregated difference feature

$$\bar{\Omega}_\theta(x) := \bar{z}(q) - p$$

and its radius, even though the implementation performs logit averaging rather than explicit feature averaging on the query side.

## H.4  STABILITY OF THE EMPIRICAL MARGIN RISK

Let $S = \{(x_i, y_i)\}_{i=1}^m$ be the original sample and let $\widehat{S}$ denote the same pairs $(x_i, y_i)$ but with the empirical margin risk evaluated using the aggregated difference features $\bar{\Omega}_\theta(x_i)$. Write

$$\Omega_i := \Omega_\theta(x_i), \qquad \widehat{\Omega}_i := \bar{\Omega}_\theta(x_i),$$

and recall that $g_\theta(x) = -\|\Omega_\theta(x)\|_2^2$. Abusing notation slightly, we write $g_\theta(\Omega)$ for the score evaluated at a difference feature $\Omega$, i.e. $g_\theta(\Omega) := -\|\Omega\|_2^2$. Assume that $g_\theta(\cdot)$ is $L_g$-Lipschitz with respect to its feature argument on the radius-$r_0$ ball in $\mathbb{R}^d$.

**Lemma 3** (Empirical margin stability). *For any $\rho > 0$,*

$$\big|\widehat{R}_{\widehat{S},\rho}(\theta) - \widehat{R}_{S,\rho}(\theta)\big| \; \leq \; \frac{L_g}{\rho m} \sum_{i=1}^m \|\widehat{\Omega}_i - \Omega_i\|_2. \tag{19}$$

*In particular,*

$$\big|\widehat{R}_{\widehat{S},\rho}(\theta) - \widehat{R}_{S,\rho}(\theta)\big| \; \leq \; \frac{L_g}{\rho} \cdot \Big(\frac{1}{m} \sum_{i=1}^m \|\widehat{\Omega}_i - \Omega_i\|_2\Big). \tag{20}$$

*Proof.* For each $i$,

$$\big|\tau_\rho\big(y_i g_\theta(\widehat{\Omega}_i)\big) - \tau_\rho\big(y_i g_\theta(\Omega_i)\big)\big| \leq \frac{1}{\rho}\big|y_i g_\theta(\widehat{\Omega}_i) - y_i g_\theta(\Omega_i)\big| = \frac{1}{\rho}\big|g_\theta(\widehat{\Omega}_i) - g_\theta(\Omega_i)\big|.$$

Since $g_\theta$ is $L_g$-Lipschitz in its feature argument,

$$\big|g_\theta(\widehat{\Omega}_i) - g_\theta(\Omega_i)\big| \leq L_g \|\widehat{\Omega}_i - \Omega_i\|_2.$$

Combining the two displays gives

$$\big|\tau_\rho\big(y_i g_\theta(\widehat{\Omega}_i)\big) - \tau_\rho\big(y_i g_\theta(\Omega_i)\big)\big| \leq \frac{L_g}{\rho}\|\widehat{\Omega}_i - \Omega_i\|_2.$$

Averaging over $i$ and using the triangle inequality yields equation 19. $\qquad\square$

For the Euclidean score class considered here we can take $L_g \leq 2r_0$, since $z \mapsto -\|z\|_2^2$ has gradient of norm $2\|z\|_2$ and is therefore $2r_0$-Lipschitz on the radius-$r_0$ ball.

## H.5  PROTOTYPE-TYPICALITY AND EMPIRICAL MARGIN REDUCTION

To make the effect of augmentation on margins more explicit, we consider a simplified linear surrogate acting on encoder-induced features for the binary pairwise problem. Recall that each pair $x = (q, p)$ is labelled $y \in \{-1, +1\}$, where $y = +1$ denotes a "same-class" (correct-prototype) pair and $y = -1$ a "different-class" pair. Let $\Omega(x) \in \mathbb{R}^d$ denote any fixed representation of pairs (for example, $\Omega_\theta(x)$ for a given encoder $\Phi_\theta$), and let $h(z) = w^\top z + b$ be a linear head on this feature space. For each pairwise label $y \in \{-1, +1\}$, let $\mu_y \in \mathbb{R}^d$ be a prototype of the corresponding pairwise class in $\Omega$-space (e.g., the conditional mean $\mu_y := \mathbb{E}[\Omega(x) \mid y]$), and write $\mu_+ := \mu_{+1}$ and $\mu_- := \mu_{-1}$.

**Lemma 4** (Prototype-aligned linear head). *Assume there exists $\alpha > 0$ with $w = \alpha(\mu_+ - \mu_-)$. Then*

$$h(\mu_+) - h(\mu_-) = \alpha\|\mu_+ - \mu_-\|_2^2 > 0. \tag{21}$$

**Lemma 5** (Aggregation toward prototypes). *For each $(x_i, y_i)$, suppose the aggregated feature satisfies*

$$\widehat{\Omega}_i = (1 - \lambda_i)\,\Omega_i + \lambda_i\,\mu_{y_i} \quad \text{for some } \lambda_i \in [0, 1], \tag{22}$$

*where $\Omega_i := \Omega(x_i)$. Then*

$$\|\widehat{\Omega}_i - \mu_{y_i}\|_2 \leq \|\Omega_i - \mu_{y_i}\|_2. \tag{23}$$

*Proof.* We have $\widehat{\Omega}_i - \mu_{y_i} = (1 - \lambda_i)(\Omega_i - \mu_{y_i})$, so $\|\widehat{\Omega}_i - \mu_{y_i}\|_2 = (1 - \lambda_i)\,\|\Omega_i - \mu_{y_i}\|_2 \leq \|\Omega_i - \mu_{y_i}\|_2$. $\qquad\square$

Define the original, prototype and aggregated margins

$$m_i^0 := y_i h(\Omega_i), \quad m_i^\mu := y_i h(\mu_{y_i}), \quad m_i^{\mathrm{agg}} := y_i h(\widehat{\Omega}_i). \tag{24}$$

**Lemma 6** (Margin interpolation). *Under the assumptions of Lemma 5, for all $i$,*

$$m_i^{\mathrm{agg}} = (1 - \lambda_i) m_i^0 + \lambda_i m_i^\mu. \tag{25}$$

*In particular, if $m_i^\mu \geq m_i^0$ for all $i$, then $m_i^{\mathrm{agg}} \geq m_i^0$ for all $i$, with strict inequality whenever $\lambda_i > 0$ and $m_i^\mu > m_i^0$.*

*Proof.* By linearity,

$$h(\widehat{\Omega}_i) = h\big((1 - \lambda_i)\Omega_i + \lambda_i \mu_{y_i}\big) = (1 - \lambda_i)h(\Omega_i) + \lambda_i h(\mu_{y_i}),$$

and multiplying by $y_i$ yields $m_i^{\mathrm{agg}} = (1 - \lambda_i)m_i^0 + \lambda_i m_i^\mu$. The monotonicity statement follows immediately. $\square$

**Proposition 2** (Empirical margin risk reduction). *Assume:*

*(i) $h$ is linear and satisfies Lemma 4;*

*(ii) For each $i$, $\widehat{\Omega}_i$ satisfies Lemma 5 for some $\lambda_i \in [0, 1]$;*

*(iii) For each $i$, $m_i^\mu \geq m_i^0$.*

*Then for any $\rho > 0$,*

$$\widehat{R}_{\widehat{S}, \rho}(h) \leq \widehat{R}_{S, \rho}(h). \tag{26}$$

*If in addition there exists $i$ with $\lambda_i > 0$, $m_i^\mu > m_i^0$, and $m_i^0 < \rho$, $m_i^{\mathrm{agg}} < \rho$, then the inequality is strict.*

*Proof.* By Lemma 6, $m_i^{\mathrm{agg}} \geq m_i^0$ for all $i$. Since $\tau_\rho$ is non-increasing, $\tau_\rho(m_i^{\mathrm{agg}}) \leq \tau_\rho(m_i^0)$ for all $i$, hence

$$\widehat{R}_{\widehat{S}, \rho}(h) = \frac{1}{m}\sum_{i=1}^m \tau_\rho(m_i^{\mathrm{agg}}) \leq \frac{1}{m}\sum_{i=1}^m \tau_\rho(m_i^0) = \widehat{R}_{S, \rho}(h).$$

For strict inequality, if for some $i$ we have $\lambda_i > 0$ and $m_i^\mu > m_i^0$, then $m_i^{\mathrm{agg}} > m_i^0$. If also $m_i^0 < \rho$ and $m_i^{\mathrm{agg}} < \rho$, then $\tau_\rho$ is strictly decreasing on $(0, \rho)$, so $\tau_\rho(m_i^{\mathrm{agg}}) < \tau_\rho(m_i^0)$ for that $i$ and the average strictly decreases. $\square$

This linear surrogate analysis explains how augmentation that moves pairwise examples toward their class-typical prototypes in $\Omega$-space tends to increase margins and reduce empirical margin risk, which is also corroborated in prior work's analysis Bai et al. (2025). In the Euclidean prototype model of the main text, a similar effect arises when query and support features are averaged within class and remain well aligned.

### H.6 PROBABILISTIC RADIUS REDUCTION WITH MULTIPLE AUGMENTATIONS

For each $i = 1, \ldots, m$ and $k = 0, \ldots, M$, let $\{\Omega_i^{(k)} \in \mathbb{R}^d\}$ be i.i.d. feature vectors with radii $R_i^{(k)} := \|\Omega_i^{(k)}\|_2$. Define

$$R_{\mathrm{orig}}^{\max} := \max_{1 \leq i \leq m} R_i^{(0)}, \qquad R_{\mathrm{aug}}^{\max} := \max_{1 \leq i \leq m} \max_{1 \leq k \leq M} R_i^{(k)}. \tag{27}$$

Define the aggregated feature and corresponding radii

$$\widehat{\Omega}_i := \frac{1}{M+1}\sum_{k=0}^M \Omega_i^{(k)}, \quad \widehat{R}_i := \|\widehat{\Omega}_i\|_2, \quad \widehat{R}^{\max} := \max_{1 \leq i \leq m} \widehat{R}_i. \tag{28}$$

**Proposition 3** (Radius reduction with $M$ augmentations). *Assume $\{\Omega_i^{(k)}\}_{i,k}$ are i.i.d. from a continuous distribution on $\mathbb{R}^d$. Then*

$$\mathbb{P}\big(\widehat{R}^{\max} < R_{\mathrm{orig}}^{\max}\big) \geq \mathbb{P}\big(R_{\mathrm{aug}}^{\max} < R_{\mathrm{orig}}^{\max}\big) = \frac{1}{M+1}. \tag{29}$$

*Proof.* Consider all $m(M+1)$ radii $\{R_i^{(k)}\}_{i,k}$, which are i.i.d. on $\mathbb{R}_+$. Let $R_{\mathrm{all}}^{\max} := \max_{i,k} R_i^{(k)}$. By continuity, this maximum is almost surely unique and, by symmetry, each of the $m(M+1)$ radii is equally likely to be the maximum. The event $\{R_{\mathrm{aug}}^{\max} < R_{\mathrm{orig}}^{\max}\}$ occurs exactly when the unique maximum lies among the $m$ original radii $\{R_i^{(0)}\}$, hence

$$\mathbb{P}\big(R_{\mathrm{aug}}^{\max} < R_{\mathrm{orig}}^{\max}\big) = \frac{m}{m(M+1)} = \frac{1}{M+1}.$$

Now fix an outcome in this event. Let $I$ be the (unique) index such that $R_I^{(0)} = R_{\mathrm{orig}}^{\max}$. Then for every $i$ and $k \geq 1$, $R_i^{(k)} < R_{\mathrm{orig}}^{\max}$, and for all $j \neq I$, $R_j^{(0)} < R_{\mathrm{orig}}^{\max}$. For $i = I$, the average of the $M+1$ vectors $\{\Omega_I^{(k)}\}$ has strictly smaller norm than the largest of them: not all $\Omega_I^{(k)}$ are equal (almost surely, by continuity), and the Euclidean norm is strictly convex, so

$$\widehat{R}_I = \left\|\frac{1}{M+1}\sum_{k=0}^{M}\Omega_I^{(k)}\right\|_2 < R_{\mathrm{orig}}^{\max}.$$

For any $j \neq I$, each $\|\Omega_j^{(k)}\|_2$ is strictly less than $R_{\mathrm{orig}}^{\max}$, hence by the triangle inequality

$$\widehat{R}_j = \left\|\frac{1}{M+1}\sum_{k=0}^{M}\Omega_j^{(k)}\right\|_2 \leq \frac{1}{M+1}\sum_{k=0}^{M}\|\Omega_j^{(k)}\|_2 < R_{\mathrm{orig}}^{\max}.$$

Thus $\widehat{R}^{\max} < R_{\mathrm{orig}}^{\max}$ on this event, so

$$\mathbb{P}\big(\widehat{R}^{\max} < R_{\mathrm{orig}}^{\max}\big) \geq \mathbb{P}\big(R_{\mathrm{aug}}^{\max} < R_{\mathrm{orig}}^{\max}\big) = \frac{1}{M+1}.$$

$\square$

### H.7 TRAINING-TIME VERSUS TEST-TIME AUGMENTATION

Let $P_{\mathrm{base}}$ and $P_{\mathrm{novel}}$ denote the base and novel pairwise distributions on $(x, y)$, and let the corresponding 0–1 pairwise risks for the encoder score $g_\theta$ be

$$R_{\mathrm{base}}(\theta) := \mathbb{P}_{(x,y)\sim P_{\mathrm{base}}}\big(y\, g_\theta(x) \leq 0\big), \qquad R_{\mathrm{novel}}(\theta) := \mathbb{P}_{(x,y)\sim P_{\mathrm{novel}}}\big(y\, g_\theta(x) \leq 0\big).$$

**Training-time augmentation on base classes.** Let

$$S_{\mathrm{base}} = \{(x_i, y_i)\}_{i=1}^{m_{\mathrm{base}}} \sim P_{\mathrm{base}}^{m_{\mathrm{base}}}. \tag{30}$$

At training time, generate $M_{\mathrm{tr}}$ augmentations per example:

$$x_i^{(0)} = x_i, \quad x_i^{(k)} = A_k^{\mathrm{tr}}(x_i),\ k = 1, \ldots, M_{\mathrm{tr}}, \tag{31}$$

with difference features $\Omega_\theta(x_i^{(k)})$. Assume a radius bound

$$\|\Omega_\theta(x_i^{(k)})\|_2 \leq r_{0,\mathrm{base}}^{\mathrm{aug}} \qquad \text{for all } i, k. \tag{—}$$

The augmented training sample is

$$S_{\mathrm{base}}^{\mathrm{tr\text{-}aug}} = \big\{(x_i^{(k)}, y_i) : i = 1, \ldots, m_{\mathrm{base}},\ k = 0, \ldots, M_{\mathrm{tr}}\big\}, \tag{32}$$

of size $m_{\mathrm{base}}(M_{\mathrm{tr}} + 1)$.

**Proposition 4** (Training-time augmentation bound on base distribution). *Under the assumptions above, there exists $C_{\mathrm{enc}} > 0$ (as in Lemma 2) such that for any $\rho > 0$ and $\delta > 0$, with probability at least $1 - \delta$ over $S_{\mathrm{base}} \sim P_{\mathrm{base}}^{m_{\mathrm{base}}}$, every encoder $\theta$ satisfies*

$$R_{\mathrm{base}}(\theta) \leq \widehat{R}_{S_{\mathrm{base}}^{\mathrm{tr\text{-}aug}},\rho}(\theta) + \frac{2C_{\mathrm{enc}}L_{\mathrm{enc}}}{\rho}\frac{r_{0,\mathrm{base}}^{\mathrm{aug}}}{\sqrt{m_{\mathrm{base}}(M_{\mathrm{tr}}+1)}} + \sqrt{\frac{\log(1/\delta)}{2m_{\mathrm{base}}(M_{\mathrm{tr}}+1)}}. \tag{33}$$

*Proof.* Apply Theorem 1 to the score class $\mathcal{G}$ on the augmented sample $S_{\text{base}}^{\text{tr-aug}}$, with sample size $m_{\text{base}}(M_{\text{tr}} + 1)$ and radius parameter $r_{0,\text{base}}^{\text{aug}}$. Lemma 2 bounds the empirical Rademacher complexity by $C_{\text{enc}}L_{\text{enc}}r_{0,\text{base}}^{\text{aug}}/\sqrt{m_{\text{base}}(M_{\text{tr}} + 1)}$, yielding equation 33. $\square$

To relate base and novel risks, assume there exists a discrepancy functional $\text{disc}_{\mathcal{G}}$ such that for all encoders $\theta$,

$$R_{\text{novel}}(\theta) \ \leq \ R_{\text{base}}(\theta) + \text{disc}_{\mathcal{G}}\big(P_{\text{base}}, P_{\text{novel}}\big). \tag{34}$$

Combining equation 34 with Proposition 4 upper-bounds $R_{\text{novel}}(\theta)$ via a term controlled by training-time augmentation on $P_{\text{base}}$ plus the discrepancy.

**Test-time augmentation on novel classes.** For the novel distribution, consider a labeled sample

$$S_{\text{novel}} = \{(x_i, y_i)\}_{i=1}^{m_{\text{novel}}} \sim P_{\text{novel}}^{m_{\text{novel}}}. \tag{35}$$

At test time, generate $M$ augmentations per example, form aggregated difference features $\widehat{\Omega}_\theta(x_i)$, and let $\widehat{S}_{\text{novel}}$ denote the sample with these aggregated features. Let $\hat{r}_{\text{novel}}$ be a radius bound for the aggregated novel features, i.e.,

$$\|\widehat{\Omega}_\theta(x_i)\|_2 \leq \hat{r}_{\text{novel}} \qquad \text{for all } i.$$

Applying Theorem 1 and Lemma 2 directly to $\widehat{S}_{\text{novel}}$ yields, with probability at least $1 - \delta$,

$$R_{\text{novel}}(\theta) \ \leq \ \widehat{R}_{\widehat{S}_{\text{novel}},\rho}(\theta) + \frac{2C_{\text{enc}}L_{\text{enc}}}{\rho} \ \frac{\hat{r}_{\text{novel}}}{\sqrt{m_{\text{novel}}}} + \sqrt{\frac{\log(1/\delta)}{2m_{\text{novel}}}}. \tag{36}$$

Proposition 3 shows that, in an idealized i.i.d. model with $M$ augmentations per example, the maximum radius of novel features decreases after augmentation with probability at least $1/(M + 1)$, which in turn tends to reduce $\hat{r}_{\text{novel}}$ relative to the original radius bound. Together with Proposition 2 (for a linear surrogate) and Lemma 3, this implies that test-time augmentation tends to both shrink the complexity term and reduce the empirical margin risk on the target distribution $P_{\text{novel}}$ itself.

In contrast, training-time augmentation mainly tightens the bound on $R_{\text{base}}(\theta)$; any bound on $R_{\text{novel}}(\theta)$ obtained via equation 34 still contains the discrepancy $\text{disc}_{\mathcal{G}}(P_{\text{base}}, P_{\text{novel}})$, which can be large when base and novel classes differ substantially. This highlights the comparative advantage of Test-time augmentation for few-shot generalisation under distribution shift.

## I  MORE IMPLEMENTATION DETAILS

### I.1  FSL MODEL TRAINING (PROTONET AND FEAT)

We follow the public FEAT repository for episodic training and evaluation, including 5-way, 1-shot meta-training; 15 queries/class for both train and evaluation; Euclidean distance for classification; Res12 as the default backbone. Key arguments (defaults shown where applicable) are exposed by train_fsl.py: *task setup* {dataset={miniImagenet, tieredImagenet, CUB, Animals}, way=5, shot=1, query=15; eval_way=5, eval_shot=1, eval_query=15}; *optimization* {max_epoch=400, episodes_per_epoch=100, num_eval_episodes=200, lr $= 10^{-4}$ (with pre-trained weights), lr_scheduler=step, step_size=20, gamma=0.2, momentum=0.9, weight_decay $= 5 \cdot 10^{-4}$}; *model* {model_class $\in$ {ProtoNet, FEAT}, backbone_class $\in$ {ConvNet, Res12}, use_euclidean (Euclidean distances), temperature=1 (ProtoNet)/64 (FEAT), lr_mul=10 for the set-to-set head}. Example FEAT commands for Res12 on tieredImagenet use lr $= 2 \cdot 10^{-4}$, step_size $\in$ {20, 40}, $\gamma = 0.5$, and temperatures temperature $= 64$, temperature2 $\in$ {64, 32}; we mirror this recipe for our FEAT runs and use the same episodic protocol for ProtoNet.

Concretely, in our re-trains we use:

- **Backbones:** Res12 (all models except CUB), ConvNet (CUB).

- **ProtoNet:** model_class = ProtoNet, Euclidean distances, max_epoch = 400, episodes_per_epoch = 100, lr = 1e-4 (pretrained), step scheduler with step_size = 20, $\gamma$ = 0.2; temperature = 1 unless otherwise tuned on validation; momentum = 0.9, weight_decay = $5 \cdot 10^{-4}$. (All other task/eval counts as above.)

- **FEAT:** model_class = FEAT, lr = 2e-4, lr_mul = 10 for the Transformer head, step scheduler with step_size in $\{20, 40\}$ and $\gamma$ = 0.5; temperature = 64, temperature2 $\in \{64, 32\}$; Euclidean distances enabled; same episodic counts as ProtoNet.

At evaluation, we sample 15,000 queries and report mean accuracy with 95% confidence intervals, matching the repository's evaluation practice.

## I.2  1S-DAug (ONE-SHOT TEST-TIME AUGMENTATION) CONFIGURATION

We implement 1S-DAug as an image-conditioned, SDEdit-style denoising pipeline with optional image-prompt adapters and a light, class-preserving geometric pre-edit ("shape tweak"). The script exposes the following arguments (defaults in ·), which we fix across all main tables unless the ablation states otherwise:

**Core diffusion/editing.**  We use stable-diffusion-v1.5 from the diffusers library as the base image generator. –noise-level $\in [0, 1] \cdot 0.7$: entry point on the diffusion trajectory (larger = more rewrite, smaller = higher faithfulness); –steps · 20: denoising steps; –cfg · 9.0: guidance scale; –seed (optional) for deterministic replication; attention slicing is enabled; VAE tiling can be toggled for large images.

**Backbone generator.**  –model · runwayml/stable-diffusion-v1-5; images are fed at $512 \times 512$ resolution. Benchmarks originally at $84 \times 84$ are upsampled before augmentation using Real-ESRGAN (Wang et al., 2021).

**Image-prompt adapter (optional).**  –ip_adapter (on/off), –ip-repo · h94/IP-Adapter, –ip_scale · 0.8 controls conditioning strength.

**Shape tweaking (geometric seed).**  Enabled via –shape_aug; parameters: –shape-aug-rotate · $20°$ (uniform in $[-R, +R]$); –shape-aug-stretch · 0.20 (anisotropic scales $s_x, s_y \in [1 - S, 1 + S]$); –shape-aug-translate · 0.025 (fraction of width/height); –shape-aug-persp · 0.12 (corner jitter fraction for a single-view projective warp). Intermediate augmented images can be saved for inspection with –save_aug.

**I/O and batching.**  Single-image mode or directory batch mode; recursive directory traversal and extension override are supported; per-image runtime and peak memory are logged (used in our efficiency table).

**Recommended ranges (used in ablations).**  Noise levels $\in \{0.25, 0.5, 0.75, 1.0\}$; shape tweaks at the defaults above or slightly weaker for fine-grained datasets. When noise is very small, diversity is limited; when very large, fidelity drops unless image conditioning is active (consistent with our qualitative/quantitative ablations).

# J  USE OF LARGE LANGUAGE MODELS

**Writing assistance.**  Yes—large language models (LLMs) were used to aid and polish writing (e.g., improving clarity, tightening tone, harmonizing notation, and converting prose to LATEX). Substantive technical content, mathematical formulations, and experimental design were authored by the authors; LLM outputs were treated as drafts and were edited for accuracy and consistency with our contributions. No text was accepted without human verification.

**Retrieval and discovery.**  Yes—LLMs were used for literature discovery and organization (e.g., surfacing related work candidates, clustering themes, and drafting citation lists). All citations included in the paper were validated by the authors against the original sources; bibliographic meta-

data and claims were cross-checked manually. LLMs were not used to generate experimental results or to fabricate evidence.

**Scope and safeguards.** LLMs were not used to generate, alter, or select experimental data; to tune hyperparameters automatically; or to produce figures or tables beyond cosmetic wording. All code and analyses were implemented and executed by the authors, and all numbers reported in the paper come from our runs. Prompts contained only non-sensitive project information and public references, and no proprietary or personally identifying data were included. Where LLM-assisted text appears (e.g., phrasing of method and related-work passages), it was reviewed for factual faithfulness and edited for technical precision.

## K    1S-DAUG VISUALIZATION

Figure 5 and Figure 6 illustrate more image generation results of our proposed method.

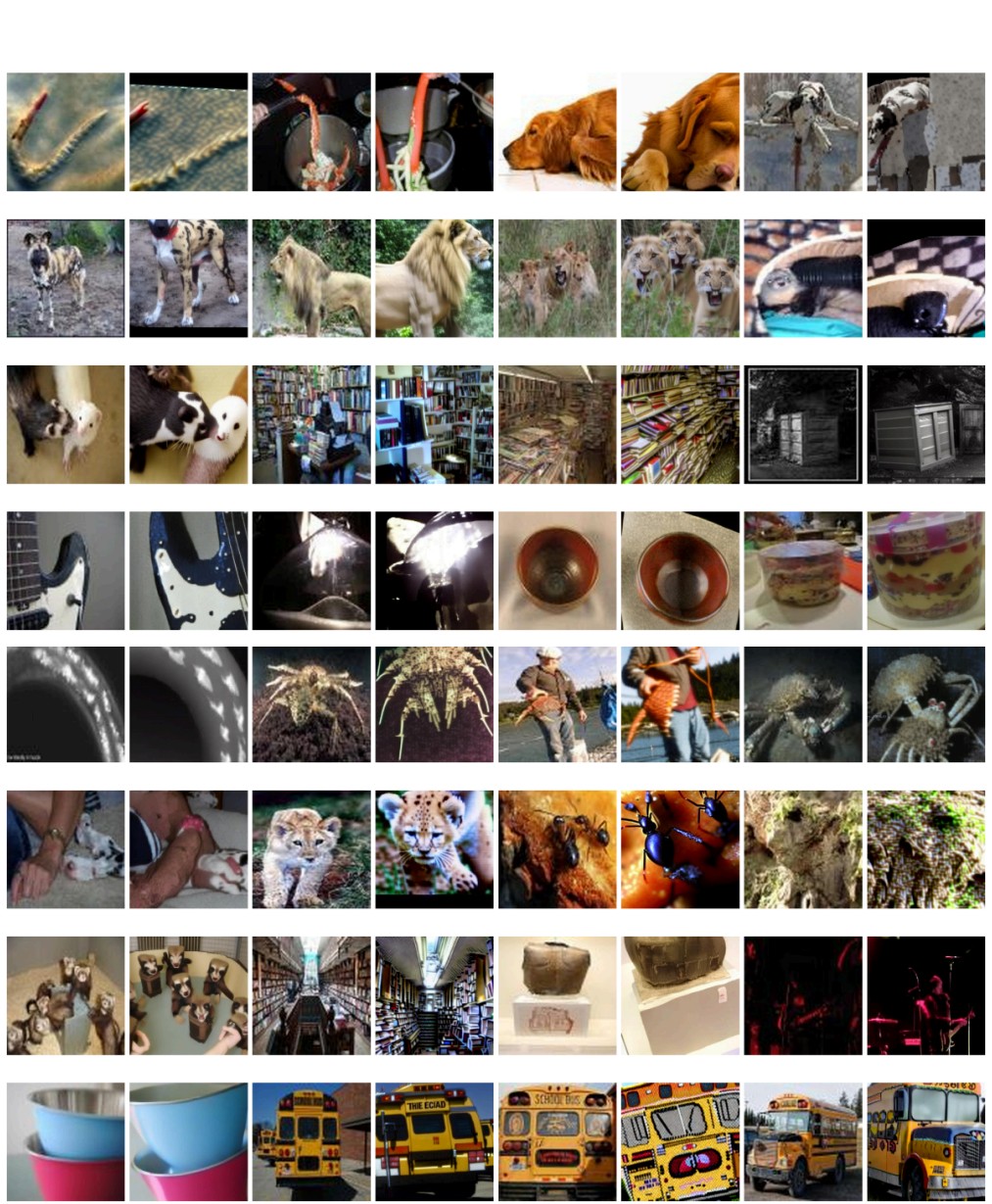

Figure 5: More qualitative results from 1S-DAug. Each pair contains the original image followed by our synthesis. All visualization pairs are random without cherry-picking.

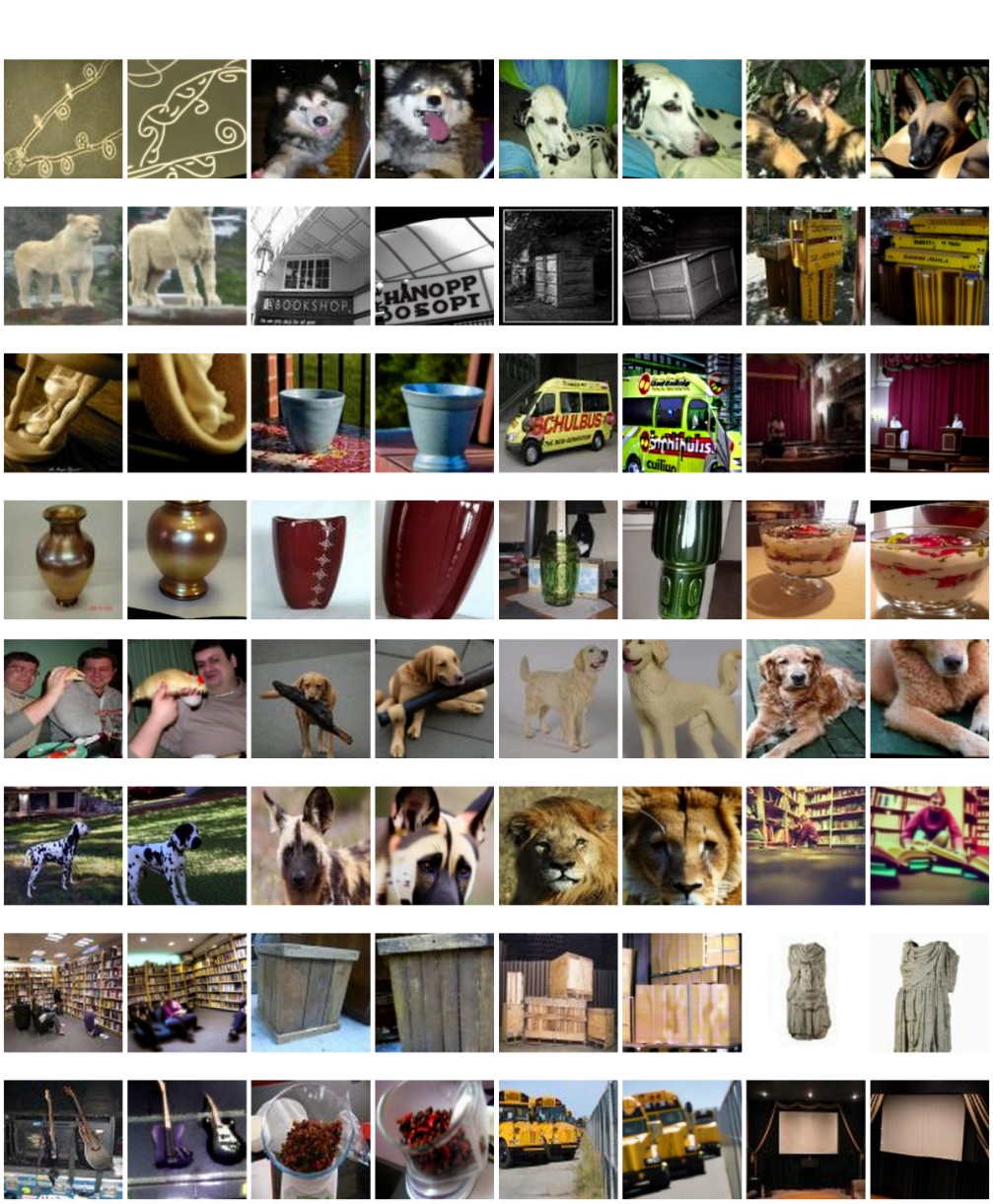

Figure 6: More qualitative results from 1S-DAug. Each pair contains the original image followed by our synthesis. All visualization pairs are random without cherry-picking.

