# OpenReview forum: "1S-DAug: One-Shot Data Augmentation for Robust Few-Shot Generalization"
_ICLR.cc/2026/Conference — ICLR 2026 Conference Withdrawn Submission_

### Official Review · Reviewer_jxZg · 2025-10-29

**Soundness:** 2
**Presentation:** 3
**Contribution:** 3
**Rating:** 2
**Confidence:** 4

**Summary:**

The authors introduce a new method to create additional samples which are different yet close enough to support few-shot image classification, especially in the lowest 1-shot setting. The approach manages to introduce variation while staying true to the core class information via a combination of geometric transformation, noise injection and targeted denoising via a ‘true’-sample-conditioned diffusion model.

**Strengths:**

**Originality & Significance:**
- The authors do a great job in outlining why the task matters, as well as how it is approached; Especially the introduced angle of looking at the value of data augmentation from the angle of ensembles adds a good layer of depth to ease the reader into the later presented method
- The idea is easy to understand, and can be employed in a model-agnostic manner (given it’s only modifying/adding to the data)
- For people with knowledge in FSL as well as diffusion models, it is also a seemingly 'obvious' way that 'should' work, and the authors do a good job of clearly outlining the ingredients that are required to make it work in practice

**Quality:**
- The work is placed well within related efforts, and the specific gap the authors tackle is clearly presented and justified

**Clarity:**
- The work is mostly well written and easy to follow, and a good mix of illustrations (esp. Figure 2) and text makes the core concepts and effect of the augmentations easy to grasp

**Weaknesses:**

- **Parts of the few-shot ‘basics’/background are misrepresented** in the current manuscript, e.g. l.166 ff: *’For inductive FSL, model parameters are frozen during inference, while the transductive set-up permits test-time adaptation.’*
$\rightarrow$ This is *not* the common definition of inductive vs. transductive, which is in contrast usually defined via the access to the unlabelled test-time samples: inductive models can very-well be finetuned on the support set (e.g. see seminal MAML) but do not (yet) have access to the unlabelled query set; whereas the union of support and query set is available from the start in transductive settings (i.e. the unlabelled information can be exploited as well)!
- **’Seminal’ choice of models**: The choice of models is limited to a toy 4-layer ConvNet and a 12-layer ResNet; Although these have been popular in the field for a long time, most recent works have demonstrated results with ViT-/Swin-variants (single or multi-scale). Given that the authors’ contribution is on the data-augmentation side and the fact that Transformer-based models do behave differently, results in this space would significantly strengthen the paper.
- **Only in-domain experiments**: Especially since the work presents a data-augmentation approach, I’d be highly interested to see how the findings transfer to cross-domain experiments (which are common in FSL)
- **Very ‘selected’, partially outdated and somewhat misleading SOTA comparison**: Newest compared method is from 2023 (Table 1); Note that newer works that use ResNet12 exist, and even some older ones perform better than the reported methods (e.g. PAL, ICCV 2021; COSOC, NeurIPS 2021, etc.);  As mentioned previously, Transformer methods would additionally strengthen the paper (e.g. FewTURE, NeurIPS 2022; HCTransformers, CVPR 2022)
- Lack in clarity why some results not reported in Table 1, as well as limited discussion of reported ones; see questions.
- Some lack in clarity around interpretation of the insights in Table 4; see questions.

**Questions:**

**TL;DR:** While I think the paper presents an interesting and straight-forward approach that could have quite a wide range of applications across FSL methods, I’m missing a number of key insights at this stage (see weaknesses above and questions below); If the authors can address (some of) these, I’m more than happy to reconsider my current rating.

For more details, please also see the weaknesses above.
My main questions relating to these:
- Can the authors provide insights into the impact of their methods on recent Transformer backbones?
- Can the authors provide insights and/or background information how their method would perform in cross-domain settings? I think this would significantly strengthen the paper, given that the domain gap usually challenges models, and it would be of interest whether augmentation in this manner could improve robustness & performance. (E.g. miniImagenet $\rightarrow$ CUB, which should be easy to evaluate; even more interesting would be the meta-dataset, but likely not feasible given limited time)
- Re Table 1: I don’t quite understand why results for some 1S-DAug methods are not reported, e.g. 1S-DAug-3 for tiered 1shot, etc?
- Re Table 4: The results show that augmenting the query as well is almost required to obtain ‘good’ results; However, doesn’t this actually challenge the authors underlying assumption of ‘better representing the true underlying distribution’ of the data where the support set is sampled from?
  How exactly are you computing the accuracy when you have multiple queries – are the query embeddings averaged (kind of query-prototypes), or are they individually classified and you aggregate the results (e.g. voting, etc.)?
  $\rightarrow$ If averaging, do you have insights as to why this helps to match better? I assume dimensions of high variation would be attenuated, and the mean moves closer to the prototype mean; but in theory, shouldn’t having more support samples be able to achieve the same goal (i.e. move the mean closer to the ‘true’ centre and therefore closer to the query)? Or does the diffusion process introduce new information/create patterns that makes matching simply easier? (Or is there simply something I’m missing?)

---

> ### Author Response · Authors · 2025-11-15
> **Responses to Reviewer jxZg**
>
> 1. **Inductive vs transductive FSL**
> We agree that under transductive FSL, all test samples are exposed; in inductive FSL, only partial query samples are given. Still, we state that our inductive setup is a standard (common) setup. This follows the convention of inductive FSL set-up descriptions among existing works (e.g., section 3.1 of the ICCV 2025 paper *Rethinking Few-Shot CLIP Benchmarks: A Critical Analysis in the Inductive Setting*), which define inductive FSL as training only with the train split. There can be alternative forms of FSL set-ups, and what we are describing is just a common practice. We will update the phrasing for better clarity.
>
> 2. **Small model and backbone/experiment scale**
> * In real life, due to the high data, computation and deployment costs, small models can actually be useful in practical settings.
>
> * We cover 4 different datasets in both 1-shot and 5-shot set-ups under 2 backbones + 2 algorithms, which require pretraining + episodic fine-tuning for each setup. Standard FSL works cover 3-4 datasets with 1 to 3 backbones. Here are a few examples published in recent years following similar experiment protocols:
>
> * MetaDiff: Meta-Learning with Conditional Diffusion for Few-Shot Learning (AAAI 2024) (conv4, res12, miniimagenet, tieredimagenet)
>
> * Prototypes-oriented Transductive Few-shot Learning with Conditional Transport (ICCV 2023) (Res18, WRN, miniImageNet, tieredImageNet, CUB, and CIFAR-FS)
>
> * Tripartite Weight-Space Ensemble for Few-Shot Class-Incremental Learning (CVPR 2025) (res18, miniImageNet, CUB200, and CIFAR100)
>
> * Generate Universal Adversarial Perturbations for Few-Shot Learning (NIPS 2024) (CIFAR-FS, mini-ImageNet, and Tiered-ImageNet, res12+res18, 5-way-5-shot experiments only on 2 datasets)
>
> * We appreciate the suggestion to include newer backbones and SOTA methods. While our experimental setup follows the common practice, we agree that adding discussion of ViT/Swin-based approaches and recent SOTA methods is also a standard practice, and will strengthen the paper. We will incorporate the results (ViT + SwinTiny).
>
> * We hope to clarify what is meant by "cross-domain experiments". For CUB/Animals, the object are in the same domain (birds/animal faces). For miniImagenet/tieredImagenet, there are already different object types (e.g., train on truck/jelly fish, test on insects). Is miniImagenet/tieredImagenet already considered cross-domain compared to birds/animal faces?
>
> 3. **Sota comparison**
> While we do not claim sota, we will explicitly state that our goal is to provide a model-agnostic test-time augmentation that can be plugged into such backbones rather than to outperform all existing specialized architectures. We will also replenish more works of comparison in our updated paper.
>
> 4. **Skipped experiment results**
> Yes. We skipped a few experiment results as we found them not so important. For example, tieredImagenet 3 augmentation case is skipped because the 1/2 augmentation copy results are already reported. Nevertheless, we acknowledge that this can be a weakness, and will fill in all results.
>
> 5. **Augment query**
> There might be a misunderstanding here. When we feed "multiple queries", we do not use the real query samples, but augmented queries generated based on one original query. We use feature averaging (see section 4) to merge the generated samples with the original samples. Having more support samples can achieve "the same goal", but having more query samples can help even further. Imagine: there are front-facing and side-facing dog reference samples in a support set. Thus, it may be easier to also have dog faces from the two different perspectives for the query, compared to having just one side-facing dog query. Importantly, our augmentation does not use any additional query (like transductive FSL does), apart from just the original query + support set. The additional query + support samples are all generated.
>
> We hope the communication can help clarify.

---

> ### Author Response · Authors · 2025-11-28
> **Experiment Results for ViT-Small and SwinTiny**
>
> Here are the experiment results for vit-small and swintiny backbones, on miniImagenet. We will also top up for all remaining datasets. In this experiment set, "(1)" represents 1 support + 1 query augmentation. We initially did not employ ViT-Small and Swin-Tiny because they commonly require additional training dataset (i.e., imagenet-1k).
>
> | Model              | 1-shot | 5-shot |
> |--------------------|:------:|:------:|
> | ViT-Small          |   71.86   |   97.24    |
> | ViT-Small + 1SDAug (1) |   80.42 (+8.56)    |   97.69    |
> | ViT-Small + 1SDAug (2) |   82.76 (+10.90)    |   -    |
> | ViT-Small + 1SDAug (3) |   83.66 (+11.80)    |   -    |
> | Swin-Tiny          |   67.32    |   96.52    |
> | Swin-Tiny + 1SDAug (1) |   75.12 (+7.80)    |   97.46    |
> | Swin-Tiny + 1SDAug (2) |   77.82 (+10.50)    |   -   |
> | Swin-Tiny + 1SDAug (3) |   78.92 (+11.60)    |   -   |
>
> Please let us know if you have any further questions, and we look forward to your reply.

---

### Official Review · Reviewer_6ZCK · 2025-10-30

**Soundness:** 2
**Presentation:** 3
**Contribution:** 2
**Rating:** 4
**Confidence:** 3

**Summary:**

This paper introduces 1S-DAug, a training-free, test-time, one-shot generative augmentation operator for few-shot classification. Given a single image, 1S-DAug (i) applies class-preserving geometric tweaks, (ii) injects a controlled amount of diffusion noise, and (iii) performs image-conditioned denoising to synthesize a handful of diverse yet class-faithful variants. Features from the original and generated views are aggregated (simple averaging) and fed to standard metric-based few-shot heads (e.g., ProtoNet/FEAT), without updating any model parameters. On four benchmarks (miniImageNet, tieredImageNet, CUB, Animals), the method yields consistent gains over the corresponding non-augmented baselines; the paper also provides an intuitive risk-decomposition showing why ensembling the augmented view can reduce error when accuracy is preserved and predictions are decorrelated, plus a margin/complexity discussion in the appendix.

**Strengths:**

- The paper is clearly written and the pipeline is easy to follow.
- Works at test time with standard few-shot backbones (ProtoNet/FEAT), requiring no fine-tuning of the classifier or the generator. This makes it practical when model parameters are fixed or restricted.
- Some theoretical backing. The risk decomposition (Eq. 12) formalizes the accuracy-vs-diversity trade-off for two-view ensembling. While simplified, this helps motivate the design.

**Weaknesses:**

1. The method is quite simple. It mainly combines basic geometric perturbations with diffusion-based denoising and feature averaging. While this simplicity is a strength in terms of clarity, it also limits novelty. In addition, the evaluation mostly compares against older baselines such as ProtoNet and FEAT, without including more recent few-shot generation or subject-level adaptation methods, such as [1, 2] and also Fsl-rectifier. These newer works would make the comparison fairer and show whether the proposed approach can still offer advantages.
2. According to Table 5, the overall improvement seems to rely heavily on the Generation Techniques. This suggests that most of the gains come from the generation process itself, while the other components contribute only marginally.
3. The method depends on a pretrained Stable Diffusion 1.5 model, which already encodes rich visual priors for most classes. It’s unclear how the approach would perform for categories that are truly novel to the generator. In such cases, the results might not be better than simple data augmentation. Conversely, with stronger modern generators (e.g., Flux or newer SD variants), the method might perform better. It would be useful to discuss how sensitive the method is to the underlying generator’s prior knowledge.

[1] ProtoAug: Provably Improving Generalization of Few-Shot Models with Synthetic Data

[2] DataDream: Few-shot Guided Dataset Generation

**Questions:**

Please refer to Weaknesses.

---

> ### Author Response · Authors · 2025-11-15
> **Responses to Reviewer 6ZCK**
>
> 1. **Theoretical grounding as strength and simplistic method**
> Our mathematical analysis takes a simple perspective (please see line 238, where we highlight the analysis is simplistic), but the diversity vs accuracy decomposition constitutes, in itself, a complete logical chain (see the 4th comment to all reviewers). Given our current focus on introducing a new, practical idea, we try not to cause information overload. Furthermore, the implementation is non-trivial. For example, as our idea is original, designing the diffusion noising and denoising pipeline itself with image conditioning requires exploration, as prior diffusion works usually use text conditioning.
>
> 2. **Alternative image generator**
> In this work, we focus on developing algorithmic ideas compatible with evolving tools. Nevertheless, we fully agree that performance still depends on the underlying generator. To avoid overclaiming, we will (i) explicitly state this as a limitation in Section 6, and (ii) add a short discussion on how stronger or weaker generators might affect performance, pointing to our partial-noise design as a way to mitigate this issue.
>
> 3. **Reliance on generation**
> Our pipeline consists of shape tweak, noise addition, denoising with image conditioning and feature averaging. Noise and shape tweak are necessary in two aspects. *First*, they help create diversity. Table 5 demonstrates performance degradation without shape tweak or noise addition, confirming their usefulness. *Second*, they reduce reliance on the pretrained diffusion model. Pure reliance on full-noise generation poses hazards. As pointed out by your third comment, the diffusion generator also cannot cover all rare object class types, and thus we should not rely on the diffusion generator's existing knowledge. That is why we do not generate from full noise, but partial noise with shape tweak, so that we keep most of the original-image information. This has been discussed at a few places, such as from line 429 to 451.
>
> 4. **Novelty and difference against other works**
> Training data augmentation is not new, and we have covered many such literatures ranging from GAN-generation to feature perturbation in our related work discussions.Besides, FSL-Rectifier requires a GAN-based image-to-image translation model given two images of similar object types (e.g., within animal faces domain). Even within the similar object domain, there are many failure cases (see figure 4 or the FUNIT official paper limitation discussions). Under the image-to-image translation protocol, each domain then requires training of a separate model (e.g., face-domain-specific; car-domain-specific; flower-domain-specific). On the other hand, our method does not make use of image-to-image translation, bringing changes based on just shape-tweak and (de)noising, which is applicable to different object types.
>
> Overall, we hope we clarify some of the questions. Thank you again.

---

### Official Review · Reviewer_7j3D · 2025-10-31

**Soundness:** 3
**Presentation:** 3
**Contribution:** 3
**Rating:** 6
**Confidence:** 3

**Summary:**

The paper introduces 1S-DAug, a method for performing few shot learning by taking the existing support set, applying geometric transforms for increasing coverage of layout and pose, and add controlled noise that distorts high-frequency content like details and artifacts while preserving the content of the images, and finally using a denoising using a diffusion model to generate augmented support samples. The model does not use the labels in any way to generate augmented samples, potentially avoiding the need for the test samples to have any similarities or distributional overlap to the training datasets.

**Strengths:**

1. The paper proposes a technically sound and exciting paradigm to few-shot learning by introducing additional support set examples at inference, instead of attempting to make the FSL network more robust during training. Test-time augmentation has been successful in many other computer vision and image processing tasks including segmentation, classification, depth prediction, etc., and this paper adds to the relatively scarce literature on essentially what constitutes test-time augmentation in FSL. The method can be applied to most base FSL methods without additional retraining, making it a strong plug-and-play method for variety of few-shot learners.
2. The experiment setup is strong, with relevant baselines and settings (5-way 1/5-shot learning). The improvements are consistent across datasets and few shot configurations.

**Weaknesses:**

1. **More baselines, comparison, and discussion**: FSL-Rectifier is a generative baseline to perform data augmentation during test-time inference. however, other methods perform use adversarial methods like adversarial GAN to generate harder examples [1], or adversarial geometric distortions [2]. Although these methods use the adversarial component during training, they can be used at inference, and their adversarial training could make performance more robust. Adding these methods to comparisons and discussions might further differentiate why the proposed method is better than or does not require sophisticated adversarial objectives for few-shot learning. The differences in augmentations considered in the paper versus those used in [2] and its implications should also be included in discussion (particularly, if [2] can be used to train an adversarial training module instead of sampling generations to produce a sharper decision boundary [1]). Another potential line of work that can add to the discussion is the role of few-shot generation methods [3] that are possibly less biased than a pretrained diffusion model on a fixed set of classes that may not be seen during inference. A very brief discussion is provided in the end of Section 6, but that could be expanded upon in the main paper or appendix. Figure 4 and "More Related Work" is a good starting point.
2. Theoretical justification is a little weak in general - Section E2 assumes that the test and augmentation sets are drawn iid from the same distribution - which is too strong of an assumption to make. An estimate of distributional distance between the test and augmentation sets and classifier error bounds being a function of this distance would be a more meaningful result to give the user an idea of how well their classifier might perform given a diffusion model and its capacity to represent the test distribution accurately. Equating the distributions just derives the effect of adding more samples into the classifier, which will lead to better decision boundaries - it is a known result, and fairly straightforward to derive.
3. Section 3 introduces the preliminaries for a reader who is already familiar with diffusion models. I would recommend introducing only the minimal notation necessary for understanding the main paper and redirecting the reader to a more detailed supplementary section or external paper for details on the DDPM/DDIM objective/learning/noise scheduling/sampling, etc. In its current form, Section 3 comes across as very hasty and does not add much value to the paper in my opinion.
4. Most diffusion models in the literature are text-conditioned, but not image conditioned, which limits the applicability of the method from existing pretrained models like StableDiffusion. The model must evaluate on text-conditioned or unconditioned diffusion models (i.e. provide ablations on the diffusion model) to see if the few-shot performance is preserved or degraded by the diffusion.
5. The section on theoretical justification is hand-wavy at best, and at best provides sufficient conditions for the risk of the augmented classifer being less than the risk of the base classifier, which are trivial in some sense. Specifically, if $\mathbb{E}[f_{\mathcal{A}}(x)l] \ge \mathbb{E}[f(x)l]$ it already means that the classifier with the augmented samples performs better than the base sample, and in that case the base sample can even be removed from the FSL. The second diversity term is always non-positive since $f$ is constrained to be $\le 1$, making only the first term an indicator of whether the risk reduces by augmentation, which is a self-fulfilling condition. A more meaningful comparison could be if the risk can somehow be proxied with some property or statistic of the augmented samples relative to the input sample which is non-trivial. I urge the authors to reduce the claims in Section 5 (since the condition for risk-reduction reduces to $\mathbb{E}[f_{\mathcal{A}}(x)l] \ge \mathbb{E}[f(x)l]$ which is better label agreement of the augmented samples - a trivial and self-fulfilling condition that also cannot be checked at inference time).
6. The reimplementations consistently score _lower_ than the numbers mentioned in the paper. Comparing with the metrics reported in the baselines immediately reduce the accuracy gap by $\sim 2\%$, weakening the claims of performance improvement of the method. Is there any specific reason why existing methods were not used?
7. The purpose of Table 3 is unclear - what is the purpose? It is known that diffusion models will take more inference steps (longer runtime). Section 6.2 can be moved to the appendix without disrupting the main claims and narrative of the paper. This section can be replaced with a plot of noise level versus performance for all the datasets and fewshot configurations and a discussion on whether the performance is dependent on the chosen $t_0$.

Minor nits:
1. Line 375 should refer to Table 4 and not Table 5.
2. Line 782 - "not suitable" -> "that are not suitable"
3. What are the shape tweaks used in the paper? I could not find a description in the main or supplementary paper.

[1] https://proceedings.neurips.cc/paper_files/paper/2018/file/4e4e53aa080247bc31d0eb4e7aeb07a0-Paper.pdf

[2] https://openaccess.thecvf.com/content_CVPRW_2020/papers/w54/Jena_MA3_Model_Agnostic_Adversarial_Augmentation_for_Few_Shot_Learning_CVPRW_2020_paper.pdf

[3] https://arxiv.org/abs/2205.15463

**Questions:**

1. In Line 101, the paper mentions "Since our augmentation is strictly one-shot during inference, it can fulfill the challenging FSL test-time augmentation requirement." -- It is not clear what the "strictly one-shot" portion is. What is a FSL test-time augmentation requirement?
2. The image-conditioned attention provides a shortcut to the denoiser - it can essentially ignore the value of $z_t$ and simply recover the image from its conditioning. How is this mechanism prevented in learning the diffusion model?
3. What is the diffusion model used in the paper? Please mention one sentence in the Evaluation Protocol, or point the reader to the excerpt in the paper where this is mentioned. The diffusion model seems to preserve the object identity well compared to a model without image conditioning (Figure 2). The details of the diffusion model are therefore important to understand the mechanisms of original-image conditioned image generation.

---

> ### Author Response · Authors · 2025-11-15
> **Responses to Reviewer 7j3D**
>
> 1. **More comparisons**
> We agree. We will include the works you mentioned into our related work in our updated paper, discussing and highlighting the difference in greater depth. We will also add more relevant works to Table 1 or appendix, for more informative comparison.
>
> 2. **What is strictly one-shot? What is a FSL test-time augmentation requirement?**
> By "strictly one-shot", we mean that our method generates additional samples based on just one copy of the original image. We do not require extra image or label input during test time. During FSL test-time 1-shot classification, the original images are the only information we have. Thus, traditional methods like GAN or text-condition diffusion are not suitable, since they cannot generate novel-class samples without more information like label conditioning or additional training samples.
>
> 3. **How is image-conditioned shortcut prevented during generation?**
> We feed the shape-tweaked image as input to the diffusion model, and only add partial noise. The image-conditioning only guides the generation direction during the partial denoising. Simply put, our mechanism prevents the shortcut through only adding in partial noise.
>
> 4. **Simplistic theoretical justification**
> In this work, we aim to illustrate the intuition of diversity vs accuracy (see comments to all reviewers, point 4). We additionally include a Rademacher complexity bound in appendix, demonstrating the contracted radius under augmentation tightens the generalization bound. We agree that the theoretical justification is simplistic, as we have also noted in our original paper (line 238). However, the justification is logically clear and coherent despite simplicity. This paper is not theory-heavy or fancy, but we focus on presenting a new idea in an easy-to-understand manner. Nevertheless, we agree that a deeper theoretical treatment would be helpful.
>
> 5. **Redundant section 3**
> We acknowledge that section may appear long. We will shorten this section and introduce only the necessary notation as suggested.
>
> 6. **Text-conditioned diffusion model**
> In this work, as you have noted, we use image conditioning instead of text conditioning. The rationale is that text conditioning may introduce confounding effect. Importantly, as we cannot assume knowledge of the novel test samples labels, text conditioning should not involve any label-related texts for test-time augmentation. We also should not rely on prior knowledge of the diffusion model. To reduce the confounding effect, we remove the text-condition prompt entirely and keep image conditioning only, since image is the only information we have during test time. Ablation studies and visualization have been conducted on image conditioning removal (table 5 and figure 2). Nevertheless, we are happy to top up experiments if we still see other necessity.
>
> 7. **Reimplementation underperformance**
> We follow the official repository of FEAT for model pretraining, episodic training and inference. All hyperparameters follow their original repo and paper details. The authors did release the official weights, but they are unfortunately even worse than our results. Nevertheless, with better base models, our method should be able to perform even better.
>
> 8. **Purpose of table 3**
> Our method's major limitation is the extra inference cost, and thus we quantify and report the extra cost, while illustrating how much extra inference time is needed for higher noise ratio. We believe this section is relatively important as it quantifies our work's major hurdle, but we are also open to discussions of alternative arrangement. Besides, ablation and visualization of different noise levels are provided in table 5 and figure 3.
>
> 9. **Which diffusion model and shape tweak**
> We use stable diffusion v1.5 (line 996). We agree that this is a relatively important piece of information, and should be stated explicitly in the main paper. We will correct this in our updated paper. Shape tweaks are composed as rotations, anisotropic stretches, translations, perspective jitters, and horizontal flips. This is specified in lines 187-188/1002-1006.
>
> 10. **Other issues**
> We agree with all feedback and will correct the issues (line 375, phrasing).
>
> Overall, we appreciate the careful understanding and constructive feedback.

---

### Author Response · Authors · 2025-11-15
**Overall Comments to All Reviewers**

Above all else, we thank everyone for the time spent.
We prioritize the following standards for our research:
1. **Novelty**
Training-time data augmentation is common. However, we are the first work to generate additional test samples based on each data point for test-time augmentation (TTA). Our method is original, and covers different object types (e.g., from hourglass to trucks), not restricted to just image-to-image translation among similar objects. There are also very few failure cases compared to the GAN-based image-to-image translation methods (See Figure 4, 5, 6).

2. **Practically meaningful**
Classification model paves the foundation for more advanced models like the large-language models, and generalization and reliability have been the major bottleneck. Unfamiliar contexts in situations like self-driving cars or even language model question-answering can pose real-life risks. Our work serves to improve the generalization via a reliable design. Besides, working from the data side, we bypass cumbersome model parameter update; this is practically meaningful especially as model training/fine-tuning grows costly nowadays.

3. **Bridge significant research gaps**
TTA is a form of ensemble, which is famously effective for reducing error rate, meaningful for out-of-distribution generalization. However, TTA is underexplored not just in few-shot learning, but in the broader context of distribution shift (See [survey paper](https://arxiv.org/pdf/2303.15361) Section 2.6). Traditional TTA like cropping/resizing are ineffective as illustrated in prior works and our own experiments (Table 5). Also, there are no other prior works reporting TTA results on mainstream few-shot benchmarks like the miniImagenet. All these are significant gaps our work partially bridges.

4. **Logically coherent and sound method design**
In statistics, if the error rate is 0.1, then 10 independent trials yield a combined error rate of 0.1^10, a vanishingly small number (line 39-41). In section 5, we only care about two things: diversity (independence) and accuracy (low error rate), because they are the only factors behind the 0.1^10 rationale. We ensure the method design is logically coherent and corroborated by ablation studies.

5. **Non-trivial and consistent experimental performance gains**
We do not pursue SOTA, but we care about performance. As reported, we have already achieved around 10% proportional accuracy improvement on popular benchmarks like the miniImagenet. The performance gain is non-trivial and consistent across 4 different datasets, again highlighting our method's practical benefit.

6. **Clear presentation**
We try our best to make the paper easy-to-read while providing thoughtful visual illustration to facilitate understanding.

7. **Faithful/responsible in reproducibility**
We provide extensive reproducibility materials, and hereby openly promise to release **all codes**, including those for ablation studies. We do not just have an organized code base, but also careful documentation for every command and arguments used.

We follow the standards, and are happy to absorb the reviewers' feedback.

---

### Author Response · Authors · 2025-12-03

We upload an improved draft mainly with:

* Experimental results under SwinTiny and ViTSmall;

* Clearer explanations and more reasonable organization (e.g., shorten diffusion preliminaries section; keep the theoretical analysis's original scope but deepen the justification; shift efficiency discussions to appendix as suggested).

Thank you again.

---

### Note · Authors · 2026-01-26

**Comment:**

Certain reviewers gave unreasonable scores without proper justification. During the rebuttal, they did not reply as well. One of them expressed interest to reconsider the score, but the commitment was not there despite us delivering the requested results. Overall, we hope the review system could be improved, giving the authors more due respect.

**Withdrawal Confirmation:**

I have read and agree with the venue's withdrawal policy on behalf of myself and my co-authors.

---

### Meta-Review · Area_Chair_QDJC · 2026-01-07

**Summary:**

In this paper, the authors propose 1S-DAug, a one-shot generative augmentation operator that synthesizes diverse yet faithful variants from just one example image at test time. The paper was reviewed by three reviewers and received mixed ratings: two rejections and one borderline acceptance. The major concerns include the technical novelty concern (training-time data augmentation is common), only in-domain experiments, and partially outdated SOTA comparisons. The authors provide detailed feedback with experimental results. But the major concerns were not fully addressed. For example, the authors mentioned that they didn't claim SOTA and their goal is not outperforming all existing specialized architectures. I think this paper might not be ready to be presented at ICLR, but it is more suitable for venues such as TMLR. Therefore, I am not able to accept this paper for now. I believe this paper needs more in-depth experiments and comparisons, even if it is not aiming at the SOTA.

**Reviewer Concerns:**

The major concerns include the technical novelty concern (training-time data augmentation is common), only in-domain experiments, and partially outdated SOTA comparisons. The authors provide detailed feedback with experimental results. But the major concerns were not fully addressed. For example, the authors mentioned that they didn't claim SOTA and their goal is not outperforming all existing specialized architectures. I believe this paper needs more in-depth experiments and comparisons, even if it is not aiming at the SOTA.

**Reviewer Scores:**

The paper was reviewed by three reviewers and received mixed ratings: two rejections and one borderline acceptance. I think the authors' responses are not able to cover all concerns, and a further revison is required before accpeting this paper.

---

### Decision · Program_Chairs · 2026-01-26

Reject